

# Holocene sea ice and paleoenvironment conditions in the Beaufort Sea (Canadian Arctic) reconstructed with lipid biomarkers

Madeleine Santos[1,2], Lisa Bröder[1], Matt O'Regan[3,4], Iván Hernández-Almeida[1,5], Tommaso Tesi[6],
Lukas Bigler[1,7], Negar Haghipour[1,8], Daniel B. Nelson[2], Michael Fritz[9], Julie Lattaud[1,2,7]

[1]Department of Earth and Planetary Sciences, ETHZ, Zurich, 8092 Switzerland

[2]Department of Environmental Science, University of Basel, Basel, 4056, Switzerland

[3]Department of Geological Sciences, Stockholm University, Stockholm, 106 91, Sweden

[4]Bolin Centre for Climate Research, – Stockholm University, Stockholm, 106 91, Sweden

[5]Past Global Changes, University of Bern, Bern, 3012, Switzerland

[6]Institute of Polar Sciences (ISP), Bologna, 40129, Bologna 40128, Italy

[7]Department of Environmental Science, Stockholm University, Stockholm, 106 91, Sweden

[8]Laboratory of Ion Beam Physics, ETHZ, Zurich, 8092, Switzerland

[9]Alfred Wegener Institute, Helmholtz Centre for Polar and Marine Research, Potsdam, 14473, Germany

*Correspondence to*: Julie Lattaud (Julie.lattaud@aces.su.se)





**Abstract** The Beaufort Sea region in the Canadian Arctic has undergone substantial sea ice loss in recent decades, primarily driven by anthropogenic climate warming. To place these changes within the context of natural climate variability, Holocene sea ice evolution and environmental conditions (sea surface temperature, salinity, terrestrial input) were reconstructed using lipid biomarkers ($IP_{25}$, and other HBIs, OH-GDGT, brGDGT, $C_{16:0}$ fatty acid, phytosterols) from two marine sediment cores collected from the Beaufort Shelf and slope, spanning the past 9.1 ka and 13.3 cal kyr BP, respectively. The Early Holocene (12-8.5 ka) is characterized by relatively higher sea surface temperature, lower salinity and no spring/summer sea ice until 8.5 ka on the Beaufort Sea slope. Around 8.5 ka, a peak in organic matter content is linked to both increased terrestrial input and primary production and may indicate increased riverine input from the Mackenzie River and terrestrial matter input from coastal erosion. Following this period, terrestrial inputs decreased throughout the Middle Holocene in both cores. A gradual increase in $IP_{25}$ and HBI-II concentrations aligns with relatively higher salinity, lower sea surface temperature and rising sea levels, and indicate the establishment of seasonal (spring) sea ice on the outer shelf around 7 ka and on the shelf around 5 ka. These patterns suggest an expansion of the sea ice cover beginning in the Middle Holocene, influenced by decreasing summer insolation. During the Late Holocene (4–1 ka), permanent sea ice conditions are inferred on the slope with a peak during the Little Ice Age. After 1 ka, seasonal sea ice conditions on the slope are observed again, alongside an increase in salinity and terrestrial input, and variable primary productivity. Similar patterns of Holocene sea ice variability have been observed across other Arctic marginal seas, highlighting a consistent response to external climate forcing. Continued warming may drive the Beaufort Sea toward predominantly ice-free conditions, resembling those inferred for the Early Holocene.



## 1. Introduction

Sea ice is a critical component of the Arctic climate system, influencing ocean–atmosphere interactions, modulating surface albedo (Kashiwase et al., 2017), regulating heat fluxes (Lake, 1967), and influencing ecosystem structure through its control on light penetration and nutrient cycling (Lannuzel et al., 2020). Its high sensitivity to temperature makes it both a driver and indicator of Arctic climate change. Since the late 1970s, satellite observations have revealed a significant decline in Arctic sea ice extent, sparking renewed interest in the mechanisms that govern sea ice variability over multiple timescales (Stroeve et al., 2012). The Canadian Beaufort Sea is a marginal sea of the western Arctic Ocean which exhibits strong seasonal and interannual variability in sea ice cover. Characterized by landfast ice on the shelf and mobile pack ice offshore, this region has experienced significant sea-ice loss in recent decades due to rising atmospheric and oceanic temperatures (Carmack et al., 2015; Comiso et al., 2008).

Understanding the natural variability of sea-ice prior to the industrial era is critical for contextualizing recent trends. Throughout the Holocene, Arctic sea ice has responded to changes in orbital forcing, ocean circulation, and ice sheet dynamics (Park et al., 2018; Stein et al., 2017). The enhanced meltwater discharge and re-routing following the retreat of the Laurentide Ice Sheet (LIS), fully deglaciated by approximately $6.7 \pm 0.4$ ka (Ullman et al., 2016), contributed to oceanographic shifts and transient cooling events, such as the Younger Dryas (~12.8–11.7 ka) (Broecker et al., 1989). Lipid biomarker records and climate simulations suggest reduced sea ice during the Early Holocene thermal maximum (11–9 ka), followed by expansion through the Middle to Late Holocene, consistent with declining summer insolation (Stranne et al., 2014; Wu et al., 2020). Numerous studies on Arctic sea ice variability have focused on a single offshore location, often neglecting the spatial extent of sea ice cover toward the coast and the migration of the marginal ice zone.

Sea-ice cover is controlled by both the atmosphere and the ocean, including salinity, sea temperature and freshwater influence, which are parameters that can be challenging to reconstruct in polar environments. Biomarker lipids and their ratios are a useful toolkit, with compound-specific hydrogen isotopes of phytoplankton biomarkers a promising tool for salinity reconstruction (e.g., Lattaud et al., 2019; Sachs et al., 2018; Weiss et al., 2019). However, in the Arctic Ocean low abundances of biomarker restricts the application of this method to the dominant biomarkers present such as palmitic acid ($C_{16:0}$ fatty acid, Sachs et al., 2018). Several proxies for sea temperature exist using microfossils (e.g., dinocyst assemblages, e.g., Richerol et al., 2008), inorganic ratios (e.g., Mg/Ca of foraminifera,



e.g., Barrientos et al., 2018; Kristjánsdóttir et al., 2007) and lipid biomarkers (e.g., Ruan et al., 2017; Varma et al.,
2024). Lipid biomarker proxies developed for reconstruction of cold water (< 15°C) temperature variations usually
include hydroxylated glycerol dialkyl glycerol tetraether (OH-GDGT) (Lü et al., 2015; Varma et al., 2024).
However, even the latest calibration of Varma et al. (2025) using a combination of OH-GDGT and isoprenoid
GDGT (isoGDGT) shows high variability at low temperature. In addition, representability of polar core-tops
sediment in the global calibration dataset is strongly biased toward the European and Russian Arctic.
This study presents a multi-proxy reconstruction of Holocene sea ice and oceanographic variability from two
sediment cores (PCB09, PCB11) collected from the Beaufort outer shelf and shelf slope. Lipid biomarkers,
including highly branched isoprenoids (HBIs), glycerol dialkyl glycerol tetraethers (GDGTs), the hydrogen isotopic
compositions of algal-derived fatty acids and terrestrial sterols, are used to reconstruct sea-ice cover, sea surface
temperatures (SSTs), salinity and terrestrial organic matter input. Additionally, a set of surface sediments is used to
assess the applicability and calibrate salinity and sea temperature proxies in sediments of the Beaufort Sea.
The primary objectives are to (1) reconstruct the spatial evolution of sea ice cover on the Beaufort Shelf throughout
the Holocene, (2) evaluate the influence of insolation, meltwater inputs, and oceanic forcing on regional sea ice
dynamics, and (3) contribute to a broader understanding of Arctic climate variability in the context of ongoing and
future climate change.
**2. Material and Methods**
**2.1. Study area**
The study focuses on the Canadian Beaufort Sea, one of the marginal seas of the Arctic Ocean (Fig. 1), bounded by
the glacially excavated Amundsen Gulf to the east, Mackenzie Trough to the west, and the Mackenzie River delta to
the south (Carmack et al., 2004). The shelf is a large estuarine setting at the interface between the Arctic Ocean and
the Mackenzie River (Omstedt et al., 1994) (Fig.1). The Mackenzie River is a significant source of freshwater to the
Beaufort Sea, with an annual water discharge of 316 km$^3$ yr$^{-1}$ (Holmes et al., 2012) and is considered the largest
Arctic river in terms of sediment flux (124–128 Mt·yr$^{-1}$) (Stein et al., 2004). At the same time, permafrost coastal
erosion adds another 8-9 Mt·yr$^{-1}$ of sediment into the Beaufort Sea, including carbon and nutrients (Wegner et al.,
2015). Surface water circulation in the Beaufort Sea is primarily characterized by the clockwise Beaufort Gyre,



which drives offshore currents towards the west and traps the majority of the Arctic Ocean's freshwater in the
Canada Basin (Serreze et al., 2006). There is also a eastward flowing shelf-break current at depths beneath 50 m,
which transports Pacific Water (coming from the Bering Strait) along the slope (Pickart, 2004). Sea ice cover on the
Beaufort Shelf north of the Mackenzie River Delta varies from year to year, but generally begins to form in mid-
October, persisting until ice break up in April-May (Fig. S1). During ice break up, an open water flaw leads occur
along the outer edge of the landfast ice allowing the formation of a spring marginal zone.

**2.2. Material**

Two sediment cores were analyzed in the study. They were recovered as part of the Permafrost Carbon on the
Beaufort Shelf (*PeCaBeau*) project during the 4[th] Leg of the 2021 CCGS Amundsen expedition (Bröder et al., 2022,
Fig.1). At station PCB09 (71.1°N, 135.1°W) at a water depth of 675 m on the Beaufort shelf slope, a piston core
(PC, length of 420 cm) and multi core (MC, 30 cm) were retrieved (Fig. 1). At station PCB11 (70.6°N, 136.0°W) on
the outer Beaufort shelf (74 m water depth) a giant gravity core (GGC, length of 290 cm) and MC (32 cm) were
recovered (Fig. 1). PCB09 is found within the modern Atlantic bottom water mass, while PCB11 lies within the
Pacific summer water (Fig. S2), the water masses were defined as in (Matsuoka et al., 2012). The core tops (0-1cm)
from 22 multicores collected during *PeCaBeau*, were used to ground truth the hydrogen isotope ratio of $C_{16:0}$ fatty
acid proxy for reconstructing salinity (Fig. S1).

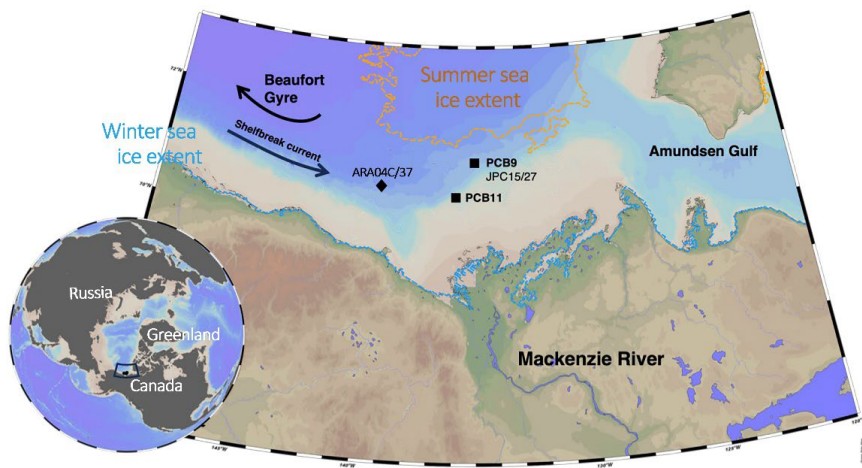


Figure 1: Combined topographic and bathymetric map of the Beaufort Shelf region (Canadian Arctic) displaying the
cores from this study as black squares (PCB09 and PCB11) and key records discussed in the text (ARA04C/37 from



Wu et al., 2020; JPC15/27 from Keigwin et al., 2018). The modern summer (orange dotted line) sea-ice extent is
shown, the winter sea ice extend follows the coastline. Map generated using Ocean Data View (Schlitzer, 2025).

**2.3. Methods**

**2.3.1. Core processing**

During the PeCaBeau expedition, all cores were scanned shipboard on a Geotek multi-sensor core logger (MSCL).
Bulk density and magnetic susceptibility were measured at a 1 cm downcore resolution on the piston and gravity cores,
(Bröder et al., 2022). The cores remained unopened and were shipped to AWI Potsdam following the expedition. They
were subsequently split in the fall/winter and the working halves sampled using 2x2 cm u-channels, before being cut
into 1-2 cm thick slices which were frozen and freeze dried.
An ITRAX XRF-core scanner was used to measure relative elemental abundances at Stockholm University, Sweden.
Measurements were performed on u-channel samples at a downcore resolution of 2 mm. Analyses were made using a
Mo tube at a voltage of 55 kV, a current of 50 mA and an exposure time of 20 s. Here we present the ratios of Ca/Ti,
reflecting detrital carbon inputs regionally elevated by meltwater delivery from either the Mackenzie or Amundsen
Gulf during deglaciation (Klotsko et al., 2019; Swärd et al., 2022; J. Wu et al., 2020). Zr/Rb was used as a proxy for
grain size variations (L. Wu et al., 2020) and Br/Cl as a proxy for marine organic matter (Wang et al., 2019).

**2.3.2. Age model**

The chronology of the piston and gravity cores (Fig. 2) were determined by $^{14}$C dating of foraminifera (n = 13,
PCB09) and bivalve shells (n = 7, PCB11) (Table S1). The MSCL data was used to stratigraphically correlate
PCB09 with JPC15/27 (Keigwin et al., 2018) allowing us to integrate existing radiocarbon ages (n=8) from this
record with our new data (n=5) (Fig. S3).
Bivalve shells were either picked from the split cores when sampling, or later from the freeze-dried sediments.
Foraminifera were picked from the >45 µm fraction of the wet-sieved samples following organic extractions.
Foraminifera samples consisted of either planktonic (*Neogloboquadrina pachyderma*), benthic, or a combination of
both in horizons when specimens were extremely rare. Care was given to pick well preserved foraminifera to avoid
age bias (Wollenburg et al., 2023). Foraminifera and mollusk samples were prepared for Accelerator Mass



Spectrometry (AMS) analyses at the Laboratory for Ion Beam Physics at ETHZ using procedures described in
(Missiaen et al., 2020) which include sieving and acid cleaning to remove impurities from the shells.
Radiocarbon-based age models were generated using the BACON package in R (Blaauw & Christen, 2011) and the
Marine20 calibration curve (Heaton et al., 2020). A reservoir age of $330 \pm 41$ years was applied to the Holocene-age
mollusc samples in PCB11 as determined by (West et al., 2022) for Pacific waters entering the Arctic Ocean in the
Chukchi Sea. For PCB09 we applied the approach used (Keigwin et al., 2018) for JPC15/27 and (J. Wu et al., 2020)
for ARA04C/37 but updated for Marine20 as described by (Lin et al., 2025). A reservoir correction of $-150 \pm 100$
years was applied to Holocene planktic foraminifera, and a larger reservoir correction ($50 \pm 100$ years) for the
bottom 4 samples (Table S1) that fall within the Younger Dryas. In our age model we also incorporate samples of
benthic foraminifera that were calibrated using a reservoir correction of $206 \pm 67$ years, determined by (West et al.,
2022) for Atlantic waters near the Chukchi Sea. Samples containing mixed planktic and benthic foraminifera were
calibrated using an average of these values ($28 \pm 85$ years).
**2.3.3. Bulk organic matter**
For the determination of total organic carbon (TOC) content and stable carbon isotope composition ($\delta^{13}$C) at the
University of Basel, about 12 mg of freeze-dried sediment was weighed into each silver capsule and 1-2 drops of
distilled water were added. The samples were exposed to fumic hydrochloric acid (HCl, 37%) in a desiccator for 24
hours to remove inorganic carbon. Samples were dried (48 h, 50 °C) and analyzed using an elemental analyser
coupled to an isotope mass spectrometer (Sercon, Integra 2). The standards used to calculate TOC was
Ethylenediaminetetraacetic acid (EDTA, Sigma Aldrich) and for $\delta^{13}$C were USGS40 (-26.389±0.042‰, IAEA),
USGS64 (-40.81±0.04, IAEA), and USGS65 (-20.29±0.04, IAEA). The analytical precision, defined as the standard
deviation of the measurement of the USGS standards for the $\delta^{13}$C sequence was ±0.03‰.
**2.3.4. Biomarkers**
5 g of homogenized freeze-dried sediment was extracted using an Energy Dispersive Guided Extraction (EDGE)
following (Lattaud, Bröder, et al., 2021). Briefly, after extraction with dichloromethane (DCM): Methanol (9:1, v/v),
the total lipid extracts (TLE) were saponified at 70 °C for two hours. The neutral phase was collected by liquid-
liquid extraction with 10 mL of hexane, three times. The leftover TLE was acidified to pH 2 and the acid phase was



recovered by liquid-liquid extraction adding 10 mL hexane:DCM (4:1, v/v), three times. The acid compounds were
methylated by adding MeOH:HCl (95:5, v/v) and heated at 70°C overnight. The methylated fatty acids were
recovered by liquid-liquid extraction (three times) with 10 mL hexane:DCM (4:1, v/v). Internal standards were
added to the neutral fraction prior to silica chromatography: 7-hexylnonadecane (7-HND, provided by S. Belt), 9-
octylheptadec-8-ene (9-OHD, provided by S. Belt), $C_{22}$ 5,16-diol (Interbioscreen), $C_{36:0}$ alkane (Sigma Aldrich) and
$C_{46}$ (Huguet et al., 2006). The neutral phase was separated into three fractions (F1, F2, and F3) through silica
column (combusted and deactivated 1%) using hexane:DCM (9:1, v/v), DCM, and DCM: Methanol (1:1, v/v).
The F1 containing HBIs was analyzed on a GC-MS (Agilent 7890-5977A) operating in Selective Ion Monitoring
(SIM) mode at the Institute of Polar Sciences (ISP), Bologna, Italy, following (Belt et al., 2014). The column used
was a J&W DB5-MS (length 30 m, id 250 μm, 0.25 μm thickness). Integrations were done in SIM mode for $IP_{25}$
(m/z = 350) and HBI IV, HBI II (m/z = 348) and HBI III (m/z = 346). Concentration of $IP_{25}$ were corrected for m/z
348 influence (4 %) and instrumental response factor. 9-OHD was used to quantify HBIs. A reference sediment
containing known amount of $IP_{25}$ was run in parallel to correct $IP_{25}$ concentration.
F3, containing the GDGTs, was filtered using a polytetrafluoroethylene filter (PTFE, 45 μm pore size) and analyzed
with high performance liquid chromatography (LC)/atmospheric pressure chemical ionization–MS on an Agilent
1260 Infinity series LC-MS according to (Hopmans et al., 2016) and following (Lattaud, De Jonge, et al., 2021).
GDGTs were quantified using the $C_{46}$ internal standard assuming the same response factor.
The F3 fraction was then sililated with bis(trimethylsilyl)trifluoroacetamid (BSTFA) (70 °C 30 min) and analysed at
the ISP for sterol concentration on a GC-MS. The $C_{22}$ 5,16 is used to quantify sterols. Specific m/z ratios have been
extracted from chromatograms in order to identify each biomarker according to their respective mass spectra.
Lipid $\delta^2H$ values were analyzed by GC-IRMS on all acid fractions having adequate compound abundance. Samples
were analyzed using splitless injection with a split/splitless inlet at 280 °C and a Restek Rtx-5MS GC column (30 m
× 0.25 mm × 0.25 μm) with helium carrier gas at 1.4 mL min⁻¹. The GC oven was held at 60°C for 1.5 min, ramped
to 140°C at 15°C min⁻¹, then to 325 °C at 4 °C min⁻¹, and held for 15 min. Column effluent was pyrolyzed at
1420°C, and $\delta^2H$ values were measured on a Thermo Delta V Plus IRMS. The $H_3^+$ factor was evaluated with each
measurement sequence to confirm stability. Values were always lower than 3 ppm mV⁻¹. Reference standards with
known isotopic compositions (Mix A7, USGS71, $C_{30:0}$ FAME; provided by Arndt Schimmelmann, Indiana
University, USA) were analyzed alongside samples to normalize values to the Vienna Standard Mean Ocean Water-



Standard Light Antarctic Precipitation (VSMOW-SLAP) scale. Standards were injected at a range of concentrations
so that peak size effects could be assessed and corrected for. Quality control samples with known $\delta^2$H values were
measured as unknowns to check precision and accuracy ($C_{16:0}$ FAME in mix $F_{8-40}$, $C_{30:0}$ FAME; Arndt
Shimmelmann), which were 4.2 ‰ or better, and 1.0 ‰ or better, respectively (n = 13-16). Final fatty acid $\delta^2$H
values of $C_{16:0}$ were corrected for added hydrogen during methylation following [Eq. 1].
$$\delta^2 H_{C16:0} = \frac{(nH_{FAME}+nH_{CH3}) \times \delta^2 H_{FAME} measured - nH_{CH3} \times \delta^2 H_{CH3}}{nH_{FAME}}$$ (1)
Where $nH_{CH3}$ = 3, $nH_{FAME}$ = 32.
**2.3.5. Biomarker ratios**
In order to describe sea ice variability in the Holocene, the $PIP_{25}$ index is used (Müller et al., 2011). The $PIP_{25}$ index
[Eq. 2] uses additional phytoplankton biomarkers (i.e. brassicasterol, dinosterol, and HBI-III) which indicate open
water conditions to compare with the abundance of $IP_{25}$ (Belt et al., 2007):
$$PIP_{25} = \frac{IP_{25}}{[IP_{25}]+[Phytoplankton\ biomarker]*c}$$ (2)
HBI-III was used in this study (Belt et al., 2015; Kolling et al., 2020; Köseoğlu et al., 2018; Smik et al., 2016) as a
reference for pelagic phytoplankton to derive $P_{III}IP_{25}$ index (afterward called $PIP_{25}$). Dinosterol was not detected in
the samples, and brassicasterol has been shown to derive mainly from terrestrial input in the region (J. Wu et al.,
2020). The c value represents the ratio of the mean concentration of $IP_{25}$ over the mean concentration of HBI-III of all
samples for each core.
Surface salinity was reconstructed using the calibration between $\delta^2$H of $C_{16:0}$ fatty acid (palmitic acid) and salinity
from the test study of Sachs et al. (2018) [Eq. 3] after testing surface sediments from multicore from the region (Fig.
S3):
$$\delta^2 H_{PA} = 4.22\ (\pm 0.6) * Salinity - 338 (\pm 15)$$ (3)
where S is salinity in practical salinity units (PSU). Based on the known calibration errors (4‰ for the $\delta^2$H
measurement), reconstructed salinity should have an associated error of $\pm$ 7 PSU.



To reconstruct sea surface temperature, hydroxylated GDGTs (OH-GDGTs) were used as the hydroxyl group in these
GDGTs is suggested to be an adaptation feature to regulate permeability in cold waters (Liu et al., 2012). In this study,
the RI-OH' [Eq. 4] and TEX-OH [Eq. 5] indexes were calculated:
$$\text{RI-OH'} = \frac{[\text{OH-GDGT-1}]+2*[\text{OH-GDGT-2}]}{[\text{OH-GDGT-0}]+[\text{OH-GDGT-1}]+[\text{OH-GDGT-2}]} \tag{4}$$
$$TEX-OH = \frac{GDGT-2+GDGT-3+Cren\ isomer}{GDGT-2+GDGT-3+Cren\ isomer+OH-GDGT-0+GDGT-1} \tag{5}$$
For the conversion from RI-OH' and TEX-OH to sea surface temperature (SST), the recent calibration of Varma et
al. (2024) is used [Eq. 6 and 7]:
$$\text{RI-OH'} = 0.04 \times SST + 0.003 \tag{6}$$
$$TEX-OH = 0.021 \times SST + 0.08 \tag{7}$$
Several organic proxies have been used to interpret terrestrial organic matter input such as branched glycerol dialkyl
glycerol tetraether (brGDGTs), long chain *n*-alkanes, and plant sterols. BrGDGTs are membrane lipids synthesized
by bacteria and are known to be ubiquitous in terrestrial environments (Schouten et al., 2013). The BIT index
(Hopmans et al., 2004) [Eq.8] is a common indicator of terrestrial input into the marine realm:
$$BIT = \frac{[BrGDGT-Ia+IIa+IIIa]}{[BrGDGT-Ia+IIa+IIIa]+Crenarcheol} \tag{8}$$
**2.3.6. Micropaleontology**
Extracted sediments were wet sieved using a 45 µm mesh. The >45 µm fraction was dried in the oven (40 °C) and
picked for foraminifera using a stereoscopic microscope. Planktonic foraminifera species are identified
(https://www.mikrotax.org/pforams/) using the morphological descriptions compiled in Microtax and counted for each
sample.




**3. Results**

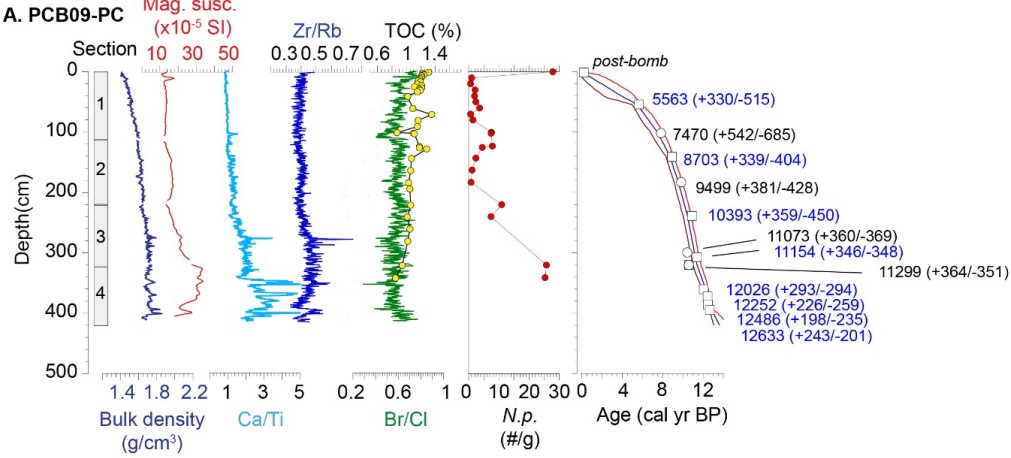

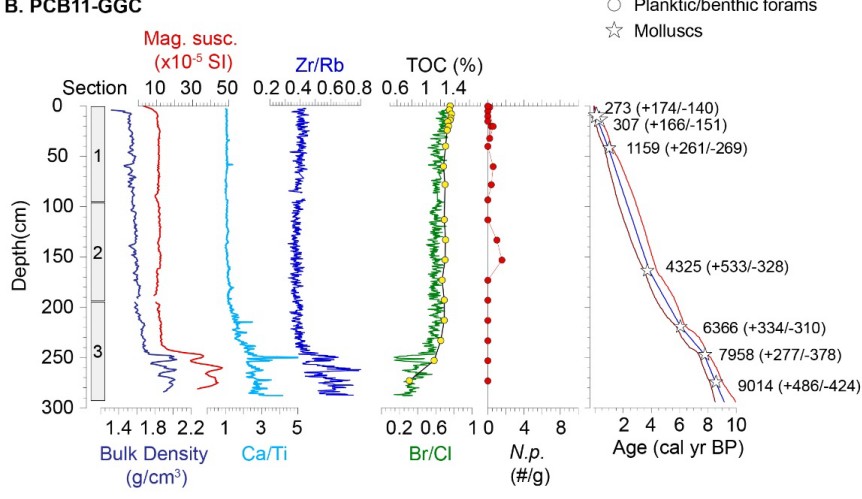


*Figure 2:* Core description for (a) PCB09 and (b) PCB11 presenting bulk density, magnetic susceptibility, X-Ray
Fluorescence (XRF) results including Ca/Ti, Zr/Rb and Br/Cl ratios. Total organic carbon content (%) and abundance
of *N. pachyderma* (*N. p.*, number/gram of sediment) as well as the age models of PCB09 and PCB11 generated using
the Bacon Rpackage (Blaauw & Christen, 2011). The red bands illustrate the 95% confidence level around the
modelled median age (blue line). Symbols indicate the calibrated age of the dated material before age-modelling. The
errors on these are lower than the symbol size. The numbers are the median modelled ages and 95% error at the
location of each sample. Blue radiocarbon ages originated from nearby core HLY13-15JPC (Keigwin et al., 2018).





### 3.1. Core chronology, lithostratigraphy and bulk organic matter

The age models of cores PCB09 (Fig. 2a) and PCB11 (Fig. 2b) show that they cover the last 13365±687 yr BP and 9115±723 yr BP, respectively. PCB11 has a mean sedimentation rate of 35 ±10 cm kyr$^{-1}$ with slightly higher sedimentation rates in the Late Holocene (< 4 ka). Conversely, PCB09 has an average sedimentation rate of 50 +/-27 cm kyr$^{-1}$, with substantially higher sedimentation rates before the Late Holocene.

The upper 300 cm of PCB09 (0 – 11 ka) display a gradual downcore increase in bulk density, reflecting porosity loss in largely homogenous silty-clay sediments (Fig 2a). Below 300 cm, slightly higher variability in the bulk density, elevated magnetic susceptibility and higher Zr/Rb all point towards a transition to slightly coarser-grained sediments. Ca/Ti tended to increase downcore becoming more variable below 105 cm (7.6 ± 0.6 cal yr BP). There was a notable stepwise increase in Ca/Ti at 345 cm (11.7 ± 0.4 ka, Fig. 2a). Higher detrital carbonate inputs are widely described in deglacial and Early Holocene sediments from the region, and generally associated with meltwater inputs from the Mackenzie River and Amundsen Gulf (Klotsko et al., 2019). Discrete peaks in Zr/Rb and bulk density, indicative of sediment coarsening, co-occured with elevated Ca/Ti ratios at depths of 276 cm (10.8 ± 0.4 cal yr BP), 345-352 (11.8 ± 0.4 cal yr BP) and 402 cm (12.8 ± 0.4 cal yr BP). These events are broadly consistent with the timing of the meltwater discharge events described by (Klotsko et al., 2019; J. Wu et al., 2020) with the oldest two associated with the Younger Dryas and pre-boreal Oscillation.

A similar pattern is seen in PCB11, where below 240 cm (7.4 ± 0.6 cal yr BP) there was an abrupt increase in bulk density, magnetic susceptibility, Zr/Rb and Ca/Ti. TOC concentrations and the Br/Cl ratio (which mirrors small scale changes in the TOC) also decreased notably through this interval (Fig. 2b, 3a). This lithologic transition post-dates the deglacial and Early Holocene detrital carbonate inputs in cores recovered from the Beaufort Sea slope (Klotsko et al., 2019). It is likely that this coarser basal facies is related to the inundation of the shelf during transgression.

Bulk sediment δ$^{13}$C of PCB09 was lowest in the Early Holocene at approximately -26.3‰ until 8.7 ± 0.4 ka, before increasing during the Middle and Late Holocene to -24.7‰. The trend in δ$^{13}$C in PCB11 is similar to PCB09 showing a steady increase over time from -26.5‰ to -25.8‰.



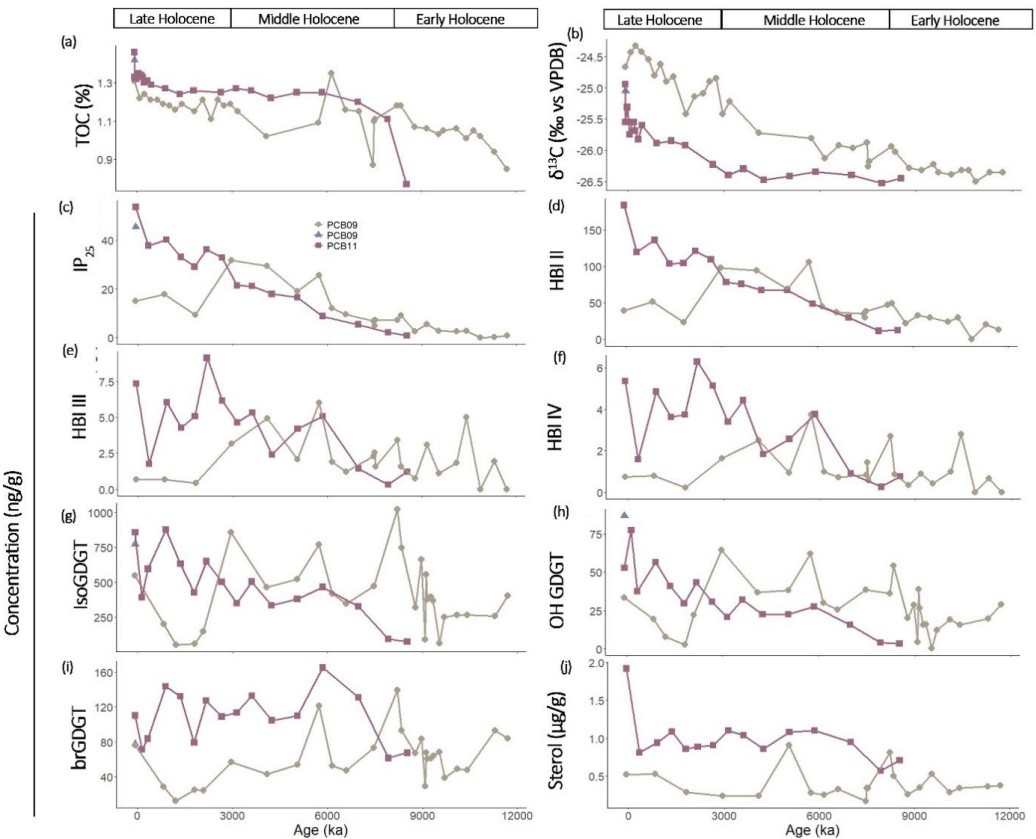


Figure 3: Bulk characteristics and biomarker concentrations (in ng/g$_{sediment}$) for core PCB09 (brown circles) and
PCB11 (red squares) with (a) total Organic Carbon (TOC), (b) $\delta^{13}$C, (c) IP$_{25}$, (d) HBI-II, (e) HBI III, (f) HBI IV, (g)
isoprenoid GDGTs (isoGDGT), (h) hydroxylated GDGTs (OH-GDGT), (i) branched GDGTs (brGDGT) and (j)
terrestrial sterols (sum of brassicasterol, stigmasterol, β-sitosterol, campesterol).
**3.2. Biomarkers**
IP$_{25}$ and HBI II (C$_{25:2}$) concentrations were generally low (< 2 ng/g) in the Early Holocene (Fig. 3c,d). IP$_{25}$ in both
cores increased throughout the Middle to Late Holocene. During the Late Holocene, IP$_{25}$ and HBI II concentrations
dropped in PCB09 around 1.8 ± 1.5 ka. Concentrations of both biomarkers were higher in PCB11 than in PCB09
after 3 ka, reaching modern values of 40 ng g$^{-1}$ and 150 ng g$^{-1}$ (IP$_{25}$ and HBI II). PIP$_{25}$ values in both cores increased
from the Early to the Middle Holocene (Fig. 4a). In PCB09, PIP$_{25}$ values decreased around 1 ka before increasing
back to modern values of 0.7.





HBI III ($C_{25:3}$) and HBI IV ($C_{25:4}$) were low in both cores with values below 8 ng g$^{-1}$ (Fig. 3e,f). Concentrations were
higher in PCB11 than in PCB09 after 4 ka
The concentration of isoGDGTs and OH-GDGTs followed a similar pattern throughout the Holocene (Fig. 3g,h).
IsoGDGTs and OH-GDGT concentrations in PCB09 were stable during the Early Holocene at around 400 ng g$^{-1}$ and
25 ng g$^{-1}$, respectively. At around 8.5 ka, the isoGDGTs and OH-GDGT amounts doubled. Throughout the Middle
Holocene, isoGDGTs and OH-GDGT concentrations were variable but above 500 ng g$^{-1}$. A drop in PCB09 to almost
below detection limits occurred between 1-1.5 ka. IsoGDGTs and OH-GDGTs in PCB11 showed a steady increase
from around 100 ng g$^{-1}$ and 10 ng g$^{-1}$, respectively, in the Early Holocene to >500 ng g$^{-1}$ and >50 ng g$^{-1}$ (Fig. 3g,h).
BrGDGTs concentrations in PCB09 were below 100 ng g$^{-1}$ throughout the cores except for peaks during the Early
and Middle Holocene at 11.2±0.3, 8.2±0.5 and 5.7±0.5 ka, the latter was also seen in PCB11 (albeit concentrations
were higher in PCB11) (Fig. 3i). Terrestrial sterol concentrations in PCB09 were relatively stable throughout the
core except for short-lived peaks at 9.5±0.4, 8.2±0.5 and 5.0±0.9 ka (Fig. 3j). In PCB11, the concentration remained
stable throughout the core after an initial increase at 6.9±0.6 ka and a peak in the surface sediment.

**3.5. Salinity, sea surface temperature (SST) and terrestrial input inferred from biomarker ratios**

Surface sediments δ$^2$H $C_{16:0}$ values from the Beaufort Sea (Fig. S3) range from -275 to -200‰, comparable with the
values obtained by the preliminary study of (Sachs et al., 2018). δ$^2$H $C_{16:0}$ values of all sediments correlate with
summer salinity ($r^2$ = 0.63, p < 0.001) and the calibration equation is the same as the one obtained by (Sachs et al.,
2018). This is to the contrary to what (Allan et al., 2023; J. Wu et al., 2025) observed in a set of surface sediments in
Baffin Bay and a downcore record of the Beaufort Sea, where no relationship with salinity is observed. This contrast
for the surface sediments could come from the different environment as Baffin Bay is a much more enclosed basin
compared to the Beaufort Sea, or that δ$^2$H $C_{16:0}$ values encompassed a too small range of salinity (31 – 33 psu).
Sea surface salinity inferred from δ$^2$H $C_{16:0}$ values at PCB09 increased from 27 psu ± 7 during the Early Holocene to
30 ± 7 psu during the Middle Holocene, and remained stable until 3 ka, increasing to 32 ± 7 psu during the Late
Holocene (Fig. 4b). Reconstructed salinity at PCB11 was more stable during the Middle Holocene and Late Holocene
(28 ± 7 psu) until an increase to 30 ± 7 psu in the last centuries. (Fig. 4b). This is in agreement with modern observation
showing lower salinities at PCB11 than around PCB09 (Fig. S2).





Two different set of SSTs were reconstructed using the OH-GDGT only (RI-OH') or a combination of OH- and
isoGDGT (TEX-OH) (Fig. 4d, Fig. S5a,b). SSTs were only reconstructed when BIT index were below 0.3 as both
calibration are sensitive to terrestrial input (Varma et al., 2025). TEX-OH reconstructed SSTs in PCB09 varied
between 7±2.6°C in the Early Holocene to 0±2.6°C between 1-1.5 ka, and reach modern values of 3±2.6°C toward
present (Fig. 4d). RI-OH' reconstructed SSTs in PCB09 give unrealistic values between 7 and 10 ka whereas PCB11
reconstructed SST is stable around 3 °C (Fig. S5b). PCB11 TEX-OH reconstructed SSTs are stable during the Early
to Middle Holocene (~5.0±2.6°C). However, reconstruction after 1 ka give large unrealistic variation (5-15°C, Fig.
S5a).
The BIT index showed a steady decrease in PCB11 throughout the Holocene and until 5 ka in PCB09 (Fig. 4c). In
PCB09, this decrease was interrupted at 1 ka with BIT index values reaching 0.4. This increase was likely due to a
relative decrease in crenarchaeol concentration (Fig. S5a) whereas brGDGT concentration did not decrease
significantly (Fig. 4i).

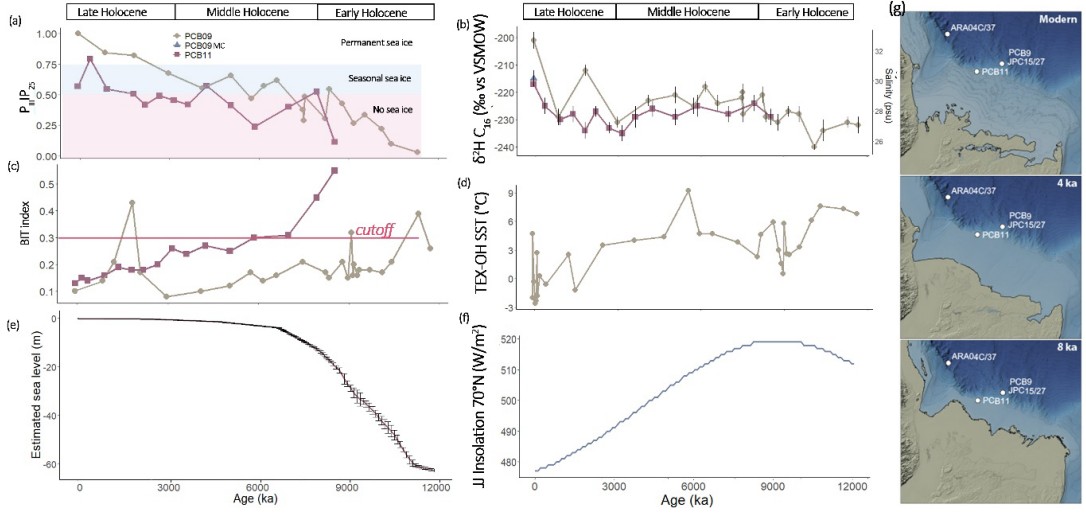


Figure 4: Reconstructed environmental parameters for PCB9 (red circles) and PCB11 (brown squares) (a) $P_{III}IP_{25}$
($PIP_{25}$ calculated with HBI III as phytoplankton biomarker), (b) $\delta^2H$ of $C_{16:0}$ fatty acid and corresponding reconstructed
salinity (Sachs et al., 2018), (c) BIT index (Hopmans et al., 2004), (d) sea surface temperature derived from TEX-OH
(Varma et al., 2024), (e), global sea level estimates derived from Lambeck et al., (2014) and (f) June-July (JJ)
insolation at 70°N (Laskar et al., 2004). Panel g) Illustrative examples of plaeoshorelines at 8 and 4 ka compared to
the modern. These were generated by adjusting the sea-level using the modern bathymetry portrayed in IBCAO V. 5
(Jakobsson et al., 2024). Relative sea-level adjustments were taken from ICE 6G_C (Peltier et al., 2015) for the grid
cell encompassing the position of PCB11. The sea-level adjustments for this location were 46 m at 8 ka and 12 m at
4 ka (Figure S7).




### 3.6. Micropaleontology

Almost all of the planktonic foraminifera (99-100% in abundance relative to other species) observed in PCB09 are *N.*
*pachyderma*, formerly *N. pachyderma sinistra*. This is expected since this species has been found to dominate polar
water masses (e.g. Eynaud 2011; Moller, Schulz, and Kucera 2013). Planktonic foraminifera are mostly absent in
PCB11, consistent with data from plankton tows indicating that planktic foraminifera are rare on the Canadian shelf
where surface waters are influenced by Mackenzie River discharge (Vilks, 1989). In PCB09, the foraminiferal shells
appear white and fragmented in sections with abundant light-colored and sand-sized ice-rafted debris and other detrital
materials (Fig. S6). Foraminifera are more abundant in samples that have relatively more mud aggregates than sand-
sized debris (Fig. 2a,b). There is almost zero accumulation rate (per mm yr$^{-1}$) of *N. pachyderma* within the shelf slope
from 10 ka.

### 4. Discussion

This study aims to reconstruct Holocene paleoenvironmental conditions in the southeastern Beaufort Sea focusing on
spatial variability between the shelf slope (> 500m water depth) and the outer shelf (<100 m water depth). By analyzing
the abundance and ratios of sea ice biomarkers (IP$_{25}$, HBI II), phytoplankton and heterotrophic archaeal productivity
markers (HBI III, HBI IV, iso- and OH-GDGT), terrestrial inputs (brGDGTs, terrestrial sterols), and reconstructed
environmental indicators (salinity, SST) this study aim to highlight spatial environmental difference between a shallow
(PCB11) and deep (PCB09) site. In the following sections, we interpret biomarker records in a chronological
framework, highlighting the dynamic relationship between freshwater inputs, ocean circulation, and sea ice conditions.

### 4.1. Deglacial to Early Holocene (12 – 8.5 ka)

The Deglacial to Early Holocene is only recorded at the shelf slope location. This period is characterized by low
concentrations of sea ice biomarkers resulting in low PIP$_{25}$ values (Fig. 3a,b, Fig. 4a). The low concentration means
that this area had some sea ice coverage during the Deglacial to Early Holocene, but the presence of HBI III and HBI
IV (Fig. 3e,f) indicate that the region was only under seasonal ice cover until spring allowing late spring/summer
open-water diatom primary production (Belt et al., 2015). Heterotrophic production in the shelf slope region during



this period is relatively low (as suggested by the presence of ammonium oxidizer Thaumarchaea-derived isoGDGTs)
but increased and peaked at 8.2 ka. During 12 – 8.5 ka, SST are elevated in comparison with the rest of the Holocene
(Fig. 4d) which coincided with peak summer insolation (Fig. 4f) (Laskar et al., 2004). The warmer surface waters
might have inhibited the development of sea ice over the Beaufort Shelf.
During the Deglacial to Early Holocene, large freshwater inputs to the Beaufort Shelf, inferred from the low
reconstructed salinity (Fig. 4b) likely originated from the decaying Laurentide Ice Sheet. Such water masses derived
from drainage regions that had undergone minimal weathering would have released low amounts of nutrients. The
influx of low-salinity freshwater may have intensified salinity-driven stratification on the shelf, reducing the upwelling
of nutrient-rich saline Pacific waters to the surface which also limited nutrient availability. This stratification and less
nutrient availability likely limited primary productivity and the presence of ammonia-oxidizers on the Beaufort Shelf.
It is important to note that sea level on the Beaufort Shelf was >60 m lower in the Early Holocene than what it is today
(Fig. 4e,g). Implying that between 10-12 ka, the Beaufort Sea was a shallow estuarine environment (Fig. 4g, Hill et
al., 1993).
The concentration of brGDGTs and terrestrial sterols in the shelf slope location during the Early Holocene peaked at
11.3 and 8.2 ka (Fig. 3i,j), which agrees with an inflow from the LIS and freshly deglaciated surfaces as seen in nearby
cores (Klotsko et al., 2019; J. Wu et al., 2020). Additionally, increased freshwater input may have transported more
detrital calcium (Ca), as indicated by elevated Ca/Ti ratios (Fig. 2a), which could have enhanced the preservation of
foraminifera by buffering the water column and limiting carbonate dissolution, in sediments along the shelf slop.
Murton et al. (2010) used optically stimulated luminescence (OSL) dating to identify two major meltwater pulses
through the Mackenzie River system between 13 and 11.7 ka and between 11.7 and 9.3 ka. This timing is supported
by sedimentary and isotopic records from the Beaufort Sea indicating a major Lake Agassiz flood route through the
Mackenzie system (Keigwin et al., 2018; Klotsko et al., 2019). These meltwater events coincide with events (11.3,
8.2 ka) in the biomarker records from this study, and one event at 10.1±0.4 ka is recorded in the reconstructed salinity
(Fig. 4b), suggesting enhanced freshwater forcing contributed to disrupted ocean circulation and increased sea ice
extent. The massive meltwater discharge from the LIS (at least ∼9000 km³) into its surrounding oceans have been the
major cause for eustatic sea level rise from 10 to 6 ka (Moran & Bryson, 1969).





**4.2. Middle to Late Holocene (8.5 – 0 ka)**

After 8.5 ka, a major cooling in SST was recorded at the slope (6 to 3 °C, Fig. 4d), sea ice biomarkers showed a steady increase in the slope and outer shelf areas (Fig. 3c,d). These trends were reflected in the $PIP_{25}$ values (Fig. 4a), where both locations started experiencing increasing sea ice cover after 8.5 ka with stable sea ice-edge or polynya conditions present at the shelf slope by 7 – 6 ka (Fig. 4a). On the outer shelf, as shown in PCB11, sea ice biomarker concentration increased along with open water diatom biomarkers. As PCB11 was very close to land before 6 ka (Fig. 4g), it is likely that the sea ice biomarkers originated from landfast ice diatoms. Between 4 -6 ka, PCB11 was likely in a "flaw-lead" position between landfast ice and sea ice, as recorded nowadys 80 km from shore (Fig. S1) ( Carmack et al., 2004). $PIP_{25}$ (Fig. 4a) in PCB11 is lower than in PCB09, indicating that stable sea ice conditions on the outer shelf were only reached after 4 ka, 2 ky later than on the slope. This delay is likely due to the position of PCB11, close to the coast and in the flaw-lead zone. In summary, during 8-3 ka, PCB11 was in a flaw-lead position or under landfast ice, enabling enhanced productivity despite increasing sea ice cover at the shelf slope.

In the Late Holocene, i.e. 3 ka to present, sea ice cover became permanent over the shelf slope as indicated by higher $PIP_{25}$ values (>0.8), increased reconstructed salinity (Fig. 4a,b), as well as a decreased amount of open water diatom (Fig. 3e,f). A sharp decrease in $PIP_{25}$ at 1.5 ka, in parallel of a steep decrease in open-water diatom biomarkers indicate a permanent sea ice cover at the shelf slope. This permanent sea ice cover occurred during a sharp decline in SST to 0 °C (Fig. 4d). Heterotrophic (ammonia oxidizer) production was inhibited, likely due to strong stratification of the water column or the presence of an ammonium-depleted water mass. This indicated a migration of the sea ice edge shoreward and a change in oceanic conditions between 1.5-0.4 ka. This period coincides with the Little Ice Age (Mann et al., 2009), when the region experienced a prolonged cold interval and the nutrient-rich Pacific water inflow was reduced (Falardeau et al., 2022). The permanent sea-ice cover was also likely restricting shelf-break upwelling of nutrient-rich deeper water (Schulze & Pickart, 2012), reducing primary productivity. At the outer shelf, seasonal sea ice conditions continued to expand and became fully established after 2 ka.



After 0.4 ka, modern sea ice conditions (Fig. S1) are established with the presence of sea ice diatoms but low
concentration of open-water diatom biomarkers at the shelf slope (Fig. 3c,d,e,f) indicating lasting ice cover. At the
outer shelf, higher sea ice and open-water diatom biomarkers are preserved in the sediment indicating polynya
conditions.

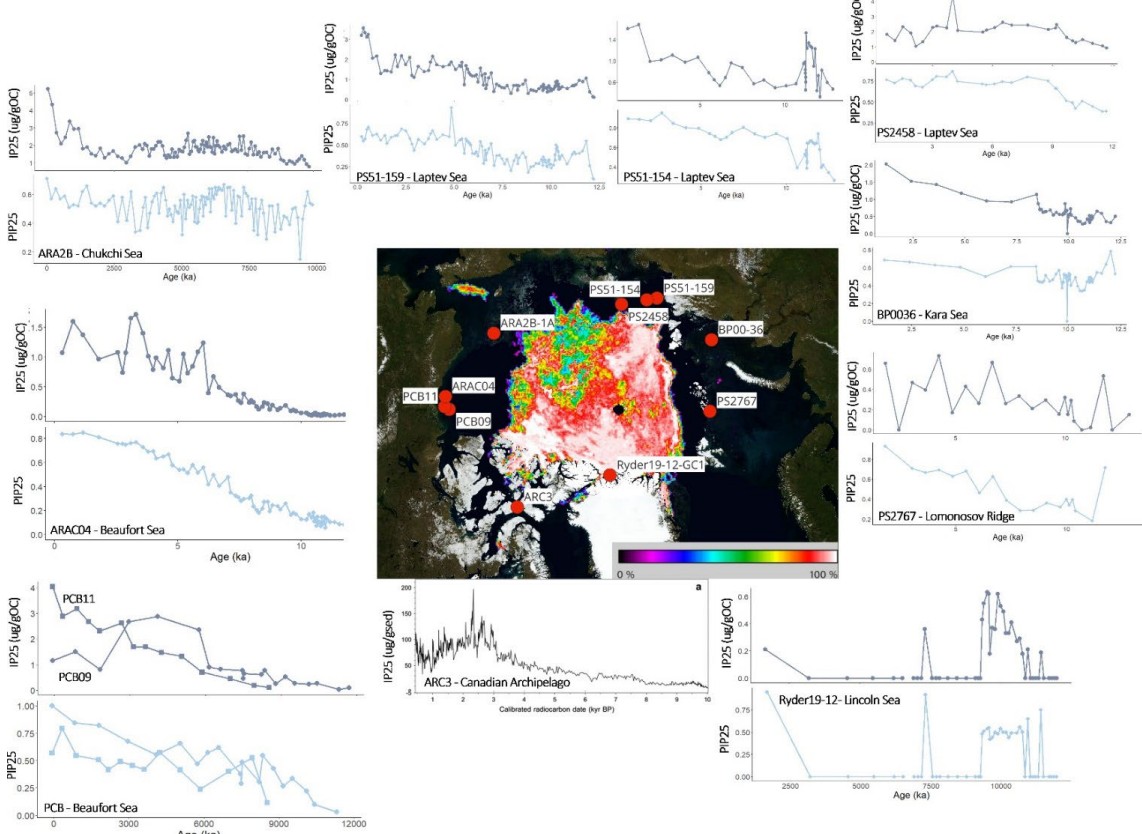

Figure 5: Arctic sea-ice records (PIP$_{25}$ and IP$_{25}$ concentration) covering the Holocene: PCB (this study), ARAC04
(Wu et al., 2020), ARA2B (Stein et al., 2017), PS51-154 and PS51-159 (Hörner et al., 2016), PS5428 (Fahl & Stein,
2012), BP0036 (Hörner et al., 2018), PS2767 (Stein & Fahl, 2012), Ryder19-12 (Detlef et al., 2023), ARC3 (Vare et
al., 2009). Satellite view from NASA (worldview worldview.earthdata.nasa.gov/), sea-ice cover for the minimum sea-
ice extend in 2021 (relative amount of sea ice as a percentage for each 12 km x 12 km from AMSR-E/AMSR2, Meier
et al., 2018).
**4.3. Comparison with other Arctic marginal seas**
Previous studies using IP$_{25}$ to reconstruct sea ice variability in Arctic marginal seas have reported largely open-water
conditions with significant freshwater influence during the Deglacial to Early Holocene.



The nearby cores JPC15 (Keigwin et al., 2018), ARAC20 (J. Wu et al., 2020) (Fig. 1) recorded similar environmental
changes (sea ice cover, freshwater input) as in PCB09 but different from those recorded in the shallow PCB11 site,
highlighting the differences between shelf break and outer shelf and the spatial variation of the polynya position. Aside
from the close by cores (Keigwin et al., 2018; Klotsko et al., 2019; J. Wu et al., 2020), other Arctic records in the
Canadian Archipelago (Vare et al., 2009), East Siberian (Dong et al., 2022), Kara (Hörner et al., 2018) , Chukchi
(Stein et al., 2017), Laptev (Fahl & Stein, 2012; Hörner et al., 2016), and Lincoln (Detlef et al., 2023) Seas and along
the Lomonosov Ridge (Stein & Fahl, 2012), report minimum sea-ice cover during the Early Holocene (centred around
10 ka) (Fig. 5). Norther Greenland(Detlef et al., 2023) and the Laptev Sea (Fahl & Stein, 2012; Hörner et al., 2016)
are the first regions to record permanent sea-ice cover after the Early Holocene minimum, around 9 ka. The Beaufort
Sea (this study, J. Wu et al., 2020) showed permanent sea-ice cover on the shelf break after 3 ka. Seasonal sea-ice
cover in the shallower region of the Laptev and Beaufort Seas (PS51-159 and PCB11) was recorded after 5 and 3 ka,
respectively. The Chukchi Sea (ARA2B) had seasonal sea-ice throughout after 8 ka, with an increase after 4.5 ka
(Stein et al., 2017). The variations in sea ice cover and primary production in the Chukchi Sea were attributed to
differences in solar insolation and variability in Pacific water inflow, which brought increased heat flux and episodic
declines in sea ice cover. In the Canadian Archipelago, a record that did not include the Early Holocene (Belt et al.,
2010; Vare et al., 2009) reported an increased sea ice cover from 7 to 3 ka. Along the Lomonosov Ridge, Stein & Fahl
(2012) described extended sea ice cover after 9 ka. Detlef et al. (2023) reconstructed sea ice conditions from a sediment
core covering the last 11 ka, showing that while the Lincoln Sea currently experiences perennial sea ice cover, it
underwent a shift to seasonal sea ice during the Early Holocene (around 10 ka) due to significantly warmer conditions.
This period of reduced sea ice cover is associated with increased marine productivity and meltwater input indicated
by biomarker and sedimentary features.
In contrast, studies using dinocyst assemblages from around the Arctic Ocean (see the review of de Vernal et al., 2013)
report constant sea ice cover for the Early to Middle Holocene with a clear decrease around 6 ka, followed by a return
to pre-6 ka conditions until an increase toward modern times. This could be due to a warm-bias in the dinocyst estimate
or a non-representative training set (de Vernal et al., 2013).
Together, many of the biomarker studies provide a consistent narrative of (spring) sea ice development during the
Holocene across the Arctic Ocean. The transition from largely open-water and freshwater-influenced conditions



during the Deglacial to Early Holocene to increasing sea ice cover from the Middle Holocene onward is a shared
feature across the Arctic shelf seas, although spatial and local variations in ice dynamics and productivity are observed
due to local freshwater input and warm current inflow.
**5. Conclusion**
Analysis of two sediment cores from the outer Beaufort Shelf and shelf slope help elucidate the region's
paleoenvironmental variability throughout the Holocene. The shelf slope had ice-free conditions and minimal sea ice
extent during the Deglacial to Early Holocene. During the Early Holocene, the Beaufort Shelf was ~60 m shallower
than today, and experienced large freshwater influxes due to the decaying LIS. The following sea level rise brought
the core sites further away from the river mouth and eroding permafrost coasts, lowering the input of terrestrial organic
matter. The insolation-based cooling recorded during the beginning of the Middle Holocene drove the increase in sea
ice cover for the Beaufort Shelf and other Arctic marginal seas. Sea-ice cover and its impact on local upwelling and
regional Pacific inflow impacted local primary production, concentrating the phytoplankton production in open-water
flaw-lead or polynya conditions. Open water conditions substantially decreased during the Late Holocene as extended
sea ice cover developed during the Little Ice Age at the shelf slope, which caused primary productivity to further
decline. This study highlights the similarities in sea ice variability across Arctic marginal seas, implying alike factors
driving sea ice variability and the impending loss of perennial sea ice condition as our modern climate approaches
thermal conditions similar or above the Early Holocene.
**Data availability**
The research data are submitted and under review on the Bolin Center database.
**Author contribution** MS - Data Curation, Formal analysis, Investigation, Writing – original draft preparation, LBr
Conceptualization, Supervision, Funding acquisition, Writing – review & editing, MO Resource, Funding acquisition,
Investigation, Writing – review & editing IH Conceptualization, Supervision, Writing – review & editing, TT
Resource, Writing – review & editing LBi Investigation, NH Resource, Writing – review & editing, DN Resource,
Writing – review & editing, MF Funding acquisition, Writing – review & editing, JL Conceptualization, Funding
acquisition, Investigation, Project administration, Supervision, Writing – review & editing



**Competing interest** The authors declare that they have no conflict of interest.
**Acknowledgments**
We thank Amundsen Science and ArcticNet for their support in preparing the cruise and the crew of the CCGS
Amundsen and the PeCaBeau team for their help during sampling. We thank the Alfred Wegener Institute for
providing the multicorer and part of the logistical support. Carina Johansson is thanked for her help with XRF analysis,
and Axel Birkholz and Thomas Kuhn for the bulk elemental and isotopic analysis. MS was funded through a SNSF-
Ambizione (PZ00P2-209012) to JL. The ship-time leading to sample acquisition for this study was funded by the
European Union H2020 as part of the EU Project ARICE (grant agreement n° 730965) with additional support from
the Swiss Polar Institute (Project number PAF-2020-004).





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
