# Peer review of "Holocene sea ice and paleoenvironment conditions in the"

_EGUsphere, 2025_

## Author Comment (AC1)

**Answer to Reviewer 1's comment:**

The goal of the manuscript "Holocene sea ice and paleoenvironment conditions in the Beaufort Sea (Canadian Arctic) reconstructed with lipid biomarkers" by Santos et al. is to fill a spatial gap in knowledge about ocean surface conditions, including sea ice, primary production, temperature, and terrestrial input, spanning the Holocene in the Beaufort Sea. The study aims to fill this gap by developing age-depth models and analyzing elemental composition, foraminifera abundance, and biomarker abundance in cores from two sites, one on the shelf (shallow), and one on the shelf slope (deeper). The study concludes that the early Holocene was warm and productive with minimal sea ice and larger inputs of terrestrial material (including organic matter and freshwater) than the late Holocene. The study also compares new and published time series and finds that these patterns are generally similar to those reconstructed elsewhere around the margins of the Arctic Ocean during the Holocene.

The central goal of this paper is important, in that quantifying the response of sea surface conditions to past periods of warmth will provide useful context for ongoing and near-future changes in the Arctic Ocean. The two study sites fill a spatial and temporal gap in data, and are based on good age constraints, especially considering the challenges with developing good age-depth models in Arctic Ocean sediments. In general appropriate methods are used, although I have a few suggestions for the authors to more clearly state the uncertainties inherent to these proxies, detailed below. The discussion sections could also be more clearly written, detailed suggestions below. Overall, the data presented here do support the conclusions. My suggestions are minor to moderate, and do not require further analysis. With some modifications to the text and figures, I recommend this manuscript for publication, as it will represent a strong and useful contribution to the literature.

We thank the reviewer for their time and positive comments. We answer the comments and indicate the planned revisions below in blue.

**Suggestions that will require moderate modifications:**

Throughout: there is some uncertainty on the ages of the time series discussed throughout the paper. It seems important to list that uncertainty when describing the timing of events. There are many examples throughout the paper, here is one: (line 361) "The concentration of brGDGTs and terrestrial sterols in the shelf slope location during the Early Holocene peaked at 11.3 and 8.2 ka". Add ± uncertainty to these ages, throughout the manuscript.

We agree with the reviewer, it is sometimes added (L267 or L268 in the results) but not in the discussion. We added the uncertainties on the modelled ages throughout the revised text.

Section 4.1 and 4.2: I'm having a hard time following whether the changes mentioned/inferred here are based on new data presented in this study or in other studies. I think most information from other studies is well cited, but there are a few spots without citations or figure callouts. I think these spots are based on data presented in this study. Can the authors add references to specific figure panels wherever data from this study are mentioned? Adding interpretive arrows to Figs 2, 3 and 4 (see suggestion below) will also help the reader follow more easily, as some of the inferences about the conditions are difficult to follow for people unfamiliar with the details of the many proxies presented here.

We added interpretive arrows in the figures as suggested and added a reference to the specific panel for all new data this study generated.

Section 4.3: I'm also having a hard time seeing in Fig. 5 some of the changes that are mentioned in the text. For example, the text states "Norther Greenland (Detlef et al., 2023) and the Laptev Sea (Fahl & Stein, 2012; Hörner et al., 2016) are the first regions to record permanent sea-ice cover after the Early Holocene minimum, around 9 ka." I think I see the pattern described here in the PIP25 time series for two of the three Laptev Sea sites (the authors could mention here that it's only the deeper Laptev Sea sites that show this pattern), but I don't see this pattern in the Northern Greenland site (in fact this site seems to have the opposite trends?). Can the authors clarify the descriptions throughout this section, so this section is easier for a reader to follow? I think adding information about the interpretations of the PIP25 ranges to Fig 5 (see Fig 5 comment) will also help.

We will guide the reader more in the revised text, and mention the deeper Laptev core recording early permanent sea-ice cover. Northern Greenland is an interesting site as IP25 production occurred during the early Holocene, indicating that there isn't a permanent sea ice cover before 9ka. The absence of IP25 and all other biomarker indicate permanent sea ice in the region without any seasonal opening of the sea-ice cover, which makes it different from the Beaufort or Laptev Sea that open in summer. Now the revised text says L465-469: " *Detlef et al. (2023) reconstructed sea ice conditions from a sediment core covering the last 11 ka, showing that while the Lincoln Sea currently experiences perennial sea ice cover ($PIP_{25} = 0$), it underwent a shift to seasonal sea ice during the Early Holocene (around 10 ka) due to significantly warmer conditions ($PIP_{25} > 0.5$). This period of reduced sea ice cover is associated with increased marine productivity and meltwater input indicated by biomarker and sedimentary facies.*"

Section 4.3: I think an important takeaway from this Arctic-wide comparison is the fact that there are a few regions that respond differently than others. This has implications for Arctic Ocean response to modern change. The authors allude to this a little bit, but a few more sentences about this conclusion would be interesting and a useful contribution. Can the authors clarify this important takeaway?

We understand that the reviewer refers to "*although spatial and local variations in ice dynamics and productivity are observed due to local freshwater input and warm current inflow*". We added one sentence to the revised manuscript L488-489: "*Evidence from areas with permanent sea ice, such as the Lincoln Sea, shows that the minimum ice cover during the Deglacial extended even into the high Arctic, offering insights into the extent of sea ice reduction during this time*".

Two suggestions about inferred Salinity:

Line 209-210: If I'm reading this sentence correctly, the ±7 psu uncertainty stems from an isotope measurement uncertainty of 4‰ and is based only on that one source of uncertainty. This estimate of uncertainty seems small, given the scatter in data points in Fig. S4b. The uncertainty on the inferred salinity measurements should also incorporate the calibration uncertainty, i.e. the uncertainty on the regression between salinity and palmitic acid isotope values. The total uncertainty reported should include both analytical and calibration uncertainty, and be propagated appropriately (i.e., typically the total uncertainty is the square root of the sum of the squares of all individual sources of uncertainty).

> The uncertainty of the salinity of +/- 7 psu already incorporates the error propagation from the calibration.

Line 289-295: Somewhere in this section, or in the discussion, it should be noted that the uncertainty in reconstructed salinity is larger than the magnitude of salinity change in the reconstruction. The authors should also address whether it is still okay to interpret the reconstructed salinity values (I think it is, as long as the caveats are made clear, and the interpretations are well supported by multiple lines of evidence)

> We will make the uncertainty clearer in this paragraph L321-322 "*It is to be noted that the uncertainty associated with the analysis and calibration reaches 7 psu, which is quite large for salinity changes during glacial-deglacial timescales.*". We agree with the reviewer that caution should be taken looking at these variations especially when compared with other records and proxies but within the cores this proxy still indicates meaningful variations. We added some information on that L332-333: "*Although uncertainties associated with reconstructed salinity area large (± 7 psu) the salinity trend between both locations agree with modern observation showing lower salinities at PCB11 than around PCB09 (Fig. S2)*"

**Suggestions that will require minor modifications:**

Line 52: lipid biomarker records where? Climate model simulations of where?

> L53: "*Lipid biomarker records and climate simulations from the Arctic*"

Line 54: rephrase to clarify: which single offshore location (or are there several studies, each of which focuses on a different offshore location)? It'd be helpful to show existing studies in a map, eg as dots on fig 1?

> L54: we removed the mention of "*single offshore locations*" to clarify. All existing studies are on Figure 1 (for the location close to our study cores) or in Figure 5 for the wider Arctic.

Line 240: the ages provided in this sentence seem very precise, given the uncertainties in the age control points. Radiocarbon labs have some information about rounding conventions for radiocarbon ages. It seems as if the authors could apply these rounding conventions to age-depth model-derived maximum core ages (e.g., https://www2.whoi.edu/site/nosams/radiocarbon-data-and-calculations/)

> We changed the ages given in L267 and 268 (to nearest 50 for ages above 10000 and nearest 10 for ages between 1000-10000)

Line 273-274: I don't understand this sentence, it doesn't describe the trends in PIP25 in either core. Remove?

> We removed this sentence, originating from an earlier draft manuscript.

Line 289: Should this be referring to Fig S4?

> Yes, we changed S3 for S4 in the revised manuscript.

Line 205-210 and line 289-295: Can the authors provide some more details and citations about which data points went into this updated isotope-salinity calibration?

> All data points from the two studies cited L317-320 (Sachs et al., 2018 and Allan et al., 2023) plus our surface sediments were added to the updated salinity calibration. All new point used are available in the supplementary. We added Allan et al., 2023 to L229.

Line 295: The salinity range quoted here (31 to 33 psu) is smaller than the range for the Baffin Bay samples shown in Fig S4B. Clarify why that's the case, or perhaps fix the quoted salinity range?

> We updated at L326 the salinity range for the study of Allan et al., 2023 (the initial text was citing mean annual salinity and not the summer salinity used in Fig S4).

Lines 304-306: seems like this could say that both cores have stable values in the middle/late Holocene?

> We agree with the reviewer and transformed the sentence to L345: "*For both cores, reconstructed SST using RI-OH' is stable around 3 °C (Fig. S5b)*".

Lines 301-305 and Figs S5a, S4d: the time series for PCB09 look different between these two figures. Perhaps this is because the PCB09-MC is plotted with the same color in Fig.4d? Can this be fixed?

> This has been fixed with the right colors.

Line 304: it's hard to see the data that support the statement that the inferred temperature approaches modern values toward present. It looks to me like the inferred temperature is highly variable in the past couple hundred years. Can this be illustrated more clearly and/or discussed differently?

> We agree with the reviewer and will add: 1) a dashed line in Fig. 4d indicating modern annual and summer temperature for reference, 2) we will discuss a bit more the temperature variations for both cores. The multicore reconstructed SST is around -1°C close to the modern annual mean surface temperature whereas the top of the piston core is about 5°C, closer to the modern surface summer mean. The revised text L335-342 now reads: "*Two different sets of SSTs were reconstructed using the OH-GDGT only (RI-OH') or a combination of OH- and isoGDGT (TEX-OH) (Fig. 4d, Fig. S4, S5a,b). SSTs were only reconstructed when the BIT index was below 0.3 (Fig. 4c) as both calibrations are sensitive to terrestrial input (Varma et al., 2025). RI-OH' in the surface sediments varies from 0.05 to 0.17 while TEX-OH varies from 0.08 to 0.32. Both indexes plot in the global calibration curves from (Varma et al., 2024) and the reconstructed SST varies from 0.9 to 4.0 °C and -0.1 to 11.6 °C, respectively. TEX-OH reconstructed SSTs in PCB09 varied between 7 ± 2.6°C in the Early Holocene, remained stable during the Middle Holocene (~3± 2.6°C), decreased to 0 ± 2.6°C between 1-1.5 ka and after which they increase to 5 ± 2.6°C, close to modern summer surface temperature (Locarnini et al., 2024).*"

Lines 309 to 312: I'm having a hard time following the explanation about the high BIT value at 1 ka in PCB09, I think perhaps because some of the 'increase/decrease' values are backwards, and because I don't see any obvious changes in the cren or brGDGT concentrations in this core at this time. Can this description be rewritten for correctness and clarity?

> This paragraph was rewritten to better follow the figures, L349-352: *"The BIT index showed a steady decrease in PCB11 throughout the Holocene and until 3 ka in PCB09 (Fig. 4c). In PCB09, this decrease was interrupted at 9 ka and at 1.5 ka with BIT index values reaching 0.3 and 0.4, respectively. The 9 ka increase was likely due to a relative decrease in crenarchaeol concentration (Fig. S5a) whereas the 1.5 ka increase was likely due to a decrease in brGDGT concentration (Fig. 4i)"*.

Lines 331-332 and Fig 2: can stratigraphic log be added to clarify the intervals that are more rich in mud vs sand? This will be useful in general, not simply for understanding the foraminifera data.

> We added the stratigraphic log to Figure 2.

Line 345: I'm confused by this statement that the HBI implies there is some sea ice, but the 'interpretation shading' in Fig 4a shows this time period is within the range of 'no sea ice'. Can this be clarified/explained in the text, or the shading in Fig 4a be modified?

> We understand the confusion. The detection of IP25 (not of PIP25) indicate the presence of sea-ice as IP25 is only produced by sea-ice diatoms (usually spring diatoms). However, when calculating PIP25 which then gives an idea of the "quality" or "extension" of the sea-ice cover, IP25 is normalized to an open-water phytoplankton marker. In summary it's two proxies for sea-ice, one for the presence of sea-ice (IP25) and the other for the extent of the sea-ice cover (PIP25). For the deglacial to early Holocene in our cores IP25 is present so there is sea-ice but PIP25 is low so there is a lot of open-water as well, so that we can then infer that there is "some sea-ice" but not a proper sea-ice marginal zone. We will make it clearer in the revised text L384-386: *"The low concentration means that this area had intermittent sea ice coverage during the Deglacial to Early Holocene, but the presence of HBI III and HBI IV (Fig. 3e,f) indicate that the region was only under seasonal ice cover until spring allowing late spring/summer open-water diatom primary production (Belt et al., 2015)"*

Line 357: can this statement about ammonia oxidizers be tied to data from the paper? If not, it sort of appears out of the blue, so should perhaps be moved or removed.

> Yes ammonia-oxidizers are the producers of iso and OH-GDGT (shown in Fig. 3) and is tied up to the changes in nutrients in the water column. This is mentioned L388-390: *"Heterotrophic production in the shelf slope region during this period is relatively low (as suggested by the presence of ammonium oxidizer Thaumarchaea-derived isoGDGTs, Schouten et al., 2013, Fig. 3g, h) but increased and peaked at 8.2 ± 0.5 ka. During 12 − 8.5 ka, SST are elevated in comparison with the rest of the Holocene (Fig. 4d) which coincided with peak 21 June insolation (Fig. 4f) (Clemens et al., 2010; Laskar et al., 2004)."*

Line 361-362: the peaks described here (and earlier in the results) are based on single data points. Can the authors provide more justification for interpreting these peaks as real?

> We agree with the reviewers that single-point changes need to be carefully interpreted. However, here a single point corresponds to a 1cm slice of sediment representing a couple of hundred years of sedimentation. The events that are reported around the meltwater pulses of the Laurentide ice sheet are only meant to last few hundred years (e.g., Wu et al., 2021).

Line 362-363: Additionally, given the interpretation of these peaks as indicating terrestrial input due to Laurentide melt, I'd expect to see the salinity decrease in the same samples. Is this the case? If not, why not?

> Yes, there is a sharp decrease in salinity around ~10 ka, similarly to the peak in terrestrial input L394-396: *"During the Deglacial to Early Holocene, large freshwater inputs to the Beaufort Shelf, inferred from the low reconstructed salinity (Fig. 4b) likely originated from the decaying Laurentide Ice Sheet."* and L409-403: *"These meltwater events coincide with events (11.3 ± 0.3, 8.2 ± 0.5 ka) in the biomarker records from this study (Fig. 3), and one event at 10.1 ± 0.4 ka is recorded in the reconstructed salinity (Fig. 4b), suggesting enhanced freshwater forcing contributed to disrupted ocean circulation and increased sea ice extent."*.

Line 364-365: it'd be helpful to see the foraminifera abundance plotted vs age for direct comparison with the other data discussed in this paragraph.

> We already have the foraminifera abundance in Fig. 2 but we will consider adding them to Figure 3 for better visibility.

Line 384-385: this interpretation is really interesting and exciting, in that it leans on modern observations and the difference between the two locations and time series. I think it'd be helpful to state more clearly that this is an interpretation (i.e. use more hedge words, such as 'may have been'), but one that is supported by multiple lines of evidence.

> We agree with the reviewer that this is an interpretation and we rewrote this paragraph L422-431 *"In contrast, at the outer shelf site PCB11, sea-ice biomarkers also increased after 8.5 ka but were accompanied by persistently high concentrations of open-water diatom markers, implying continued seasonal sea-ice and productive flaw-lead conditions, i.e., an open-water or newly formed sea-ice zone between landfast ice and sea ice. The flaw-lead today occurs about 80 km away from shore (Fig. S1) (Carmack et al., 2004). The proximity of PCB11 to the coastline before 6 ka (Fig. 4g) likely favored landfast-ice diatom assemblages and greater sensitivity to freshwater discharge. Only after 4 ± 0.5 ka did PCB11 reach PIP$_{25}$ values comparable to the slope, indicating a delayed transition to stable seasonal sea ice, approximately 2 kyr later than at PCB09. Thus, while both sites record a Middle-Holocene trend toward increasing sea-ice cover, the slope experienced earlier stabilization and reduced productivity linked to offshore cooling and stratification, whereas the outer shelf remained a dynamic, seasonally open-water environment likely sustained by coastal flaw-lead formation, and strong riverine influence."*.

Can the authors describe what a 'flaw lead' is?

Yes, a flaw lead is a sea-ice feature: an elongated zone of open water, or newly formed ice that develops between the landfast ice and the mobile sea-ice. We illustrated this situation (which occurs during modern time) in Fig. S1. We added one line defining this particular feature in the revised text L424 "*was likely in a "flaw-lead" position, i.e., at an open-water or newly formed sea-ice zone between landfast ice and sea ice*".

Line 450: remove 'during the Little Ice Age', as it is redundant with the 'during the late Holocene' statement earlier in the sentence.

We removed it accordingly.

**Suggestions for figures:**

Figure 1: define 'modern', provide citation for sea ice data source, could zoom in on inset map of whole arctic ocean and show dots of previous publications examining past sea ice cover, which are referred to in the introduction

We will update the legend of Figure 1 to include the source of the sea-ice margin and the meaning of modern (here it is the sea-ice extent for 2021).

We decided to keep Figure 1 simple, we zoom out of the study zone and into the Arctic in Figure 5, where we also remind the readers of the location of our study cores.

Figs 2, 3, and Fig 4: It'd help to add interpretive arrows on each panel, e.g. panel 3h would have an arrow pointing up labeled "increased terrestrial contribution", or something like that. Can the authors add an interpretive arrow to each panel in these figures?

We will add an interpretive arrow next to each revised panel (see below).

[Figure]

Figure 3

Fig 4 a-f: I think it'd be easier to see how these various time series align if they're arranged in a single stack plot, perhaps with some dashed vertical lines every 1 or 2 kyr.

We arranged the panels vertically in the revised figure 4 (see below)

[Figure]

Fig 4d: what is the uncertainty in inferred values using this calibration? It'd be helpful to show a vertical line that's the uncertainty, or some shading around the datapoints indicating the uncertainty.

> The uncertainty from TEX-OH sea surface temperature reconstruction is +/-2.6C, we added this in panel d.

Fig 4f: It's most appropriate to compare with peak annual insolation, as this is the forcing that the climate system responds to. 21 June insolation is in phase with peak annual insolation (see Clemens et al 2010 Fig. 6 doi.org/10.1029/2010PA001926 for an explanation about this), so I'd suggest modifying this to plot 21 June insolation instead of mean June and July insolation, as the most appropriate point of comparison for the time series.

> We thank the reviewer for this suggestion and changed panel f for 21 June insolation in the revised figure 4. It was calculated from Laskar 2004 orbital parameters.

Fig 5. Are the interpretation cutoffs displayed using shading in Fig 4a applicable to all of the PIP25 time series shown in Fig 5? If so, it could be helpful to display those shaded regions in

these figures as well. If not, it seems important to explain that they are not, and why they are not.

The cutoffs are valid for all panels so we will add shading areas in revised Figure 5 (see below).

[Figure]

Fig S6: can some arrow annotations be added to this figure to highlight the different features of interest in these images?

*We think that figure S6 has enough description in the legend: "has abundant sandy detrital clasts with fragmented and sparse foraminifera shells", "abundant foraminifera shells".*

Table S1 should also include information about the material dated.

We will add this information to the revised table S1.

---

## Author Comment (AC2)

**Response to Reviewer 3's comment:**

In this study, the authors report a multiproxy dataset analyzed on two sediment cores from the shelf slope and outer shelf of the Beaufort Sea. These records span the Holocene, and indicate that—similar to other proxy-based reconstructions in other sectors of the Arctic Ocean—the sea-ice extent in the Beaufort Sea has been increasing since the early Holocene.

The manuscript is well-written, and the dataset is a welcome addition to the literature. I applaud the authors for producing such a diverse dataset. The main conclusion about the Holocene trend in sea ice is supported by the IP25 and PIP25 data. However, I am not fully convinced yet by some of the discussion based on individual or few data points, and by some interpretations of the biomarker data. In addition, I feel that the organization of the figures and parts of the discussion can be further improved. I hope my suggestions and comments below will be helpful in revising the manuscript to improve its clarity and accessibility. Overall, I recommend **moderate to major revisions**.

We thank the reviewer for their time and constructive comments. We answer the comments and indicate the planned revisions below in blue.

General comments

[1] Interpretation of biomarker abundance

The biomarker data are presented as concentration (ng/g sediment extracted). In a region where large shifts in sedimentation regime occur, the concentration may reflect not only biomarker production but also dilution by clastic material and oxic degradation. For example, is it possible that the downcore decrease in almost all biomarkers in Figure 3 reflects oxic degradation post burial? TOC also shows such a downcore decrease; normalizing the biomarker data using TOC, as some workers prefer, would probably eliminate the Holocene "increase" in other biomarker data (e.g., HBIs, isoGDGTs, and OH GDGTs). I understand that it is not always possible to fully tease apart these factors, but I encourage the authors to discuss this possibility and, if applicable, caveat their discussion to acknowledge the uncertainty of these data.

We appreciate the reviewer's comment regarding the potential influence of dilution by clastic input and post-depositional oxic degradation affecting biomarker concentrations when expressed per gram of sediment. We agree that such effects can, in some settings, obscure original variations in biomarker production, and we acknowledge that normalization to organic carbon (OC) content is one way to reduce the influence of variable mineral dilution. In our cores, however, TOC contents are relatively stable downcore (ranging only from 0.9 to 1.3% for PCB09 and from 1.1 to 1.3% for PCB11 after the initial increase; Figure 3a). Given this limited variation, normalization to TOC would have only a minor effect on the relative biomarker profiles (see figures below for selected biomarkers). Moreover, the different downcore behaviors observed among biomarker groups (Figure 3) suggest that their trends are not governed by a single control such as degradation or dilution, but rather reflect differences in source or preservation processes. We have added a statement in the revised results acknowledging that post-depositional degradation may contribute to some of the observed downcore decrease, but that the overall patterns are unlikely to be an artifact of dilution or TOC normalization. L313-317 "*Although biomarker concentrations expressed per gram of sediment may be influenced by mineral dilution or post-depositional degradation,*

*TOC contents in our cores varied only slightly (0.9-1.3 % in PCB09 and 1.1-1.3 % in PCB11; Figure 3a). Such limited variability indicates that TOC normalization would not substantially alter the observed trends. The distinct downcore patterns among lipid biomarker classes therefore likely reflect differences in source input and preservation rather than a uniform effect of degradation or dilution.*

*.".*

[Figure]

[2] BrGDGTs as terrestrial indicator

The authors interpret brGDGT concentration as terrestrial input. However, recent studies have proposed in situ production of brGDGTs (e.g., Xiao et al., 2016, Biogeosciences). In particular, based on the IIIa/IIa ratios, Singh et al. (2025, OG) suggested that brGDGTs in the central Arctic are of marine origin and transported from the shelf. Might this explain the differing temporal patterns in the concentrations of brGDGTs and sterols? I suggest that the authors first discuss the origin of brGDGTs in the Beaufort Sea before interpreting brGDGT concentration as terrestrial input.

*We agree that brGDGT can be produced in situ in the marine environment as was described also by Peterse et al., 2009, in Svalbard or Sinninghe Damsté et al. (2016) in the Berau Delta (just to cite these two additional studies). As described in the latter, a good indicator for in situ production is the use of #ring_{tetra} (rather than IIIa/IIa) in coastal sediments. In our records, #ring_{tetra} is always below 0.7, which has been defined by Sinninghe Damsté et al. (2016) as*

the cutoff values for in situ production in most coastal shelves. We added this information L248-251 "*In addition, in situ marine production of brGDGT can occur in coastal sediments between 50 and 300 m water depth (Peterse et al., 2009; Sinninghe Damsté, 2016). To assess the potential for brGDGT to be in situ produced we calculated #ring$_{tetra}$ [Eq 9]*" in the methods and in the results L306-307 "*BIT values varied from 0.1 to 0.4 in PCB09 and from 0.1 and 0.5 in PCB11. #ring$_{tetra}$ values were always < 0.7 for both cores.*".

[3] Discussion not supported by the temporal resolution and precision of records

The temporal resolution of proxy records is relatively low, and there are large fluctuations in the data. In some parts of the manuscript, a single data point is interpreted as representing a climate event (e.g., "A sharp decrease in PIP25 at 1.5 ka…"). Given the temporal resolution of the proxy records (e.g., 15 HBI III data points over 9 ka, corresponding to ~600 years between data points) and the overall noise in the data, I am not convinced that the change between two data points (1.5 ka to 0.4 ka) can be robustly linked to the Little Ice Age—especially considering age-model uncertainty.

We rephrased this interpretation in the revised text, see for example L439-440 "*These changes are broadly consistent with the timing of the regional cooling associated with the Little Ice Age (Mann et al., 2009), though the resolution of the biomarker record does not allow precise attribution to centennial-scale events.*".

More generally, proxy records are subject to non-climatic noise from sedimentation and analytical error; unless demonstrated otherwise, interpreting absolute point-by-point values seems overconfident. It would be more robust to interpret overall trends or time-slice means instead of individual fluctuations. This can be achieved by applying a running average, smoothed series, or trend lines for defined intervals (e.g., Early, Middle, and Late Holocene).

We agree with the reviewer and we generally aim to not interpret single data points if they are not part of a general trend. However, when the changes between two data points are particularly large and reflected by several other proxies (and not readily explainable by changes in deposition – sedimentation rate, or degradation – or TOC) we at least discuss the transition without overstating their significance. In addition, we considered presenting running average for this dataset but we believe we have too low resolution for this as this study focused on presenting many different proxies rather than focusing on obtaining high resolution profile for 1 proxy (see example of running average for our dataset with higher resolution, TOC, d13C).

[Figure]

[Figure]

[4] Overall clarity and accessibility

At several points, it was challenging to follow the authors' reasoning, as it was unclear how some conclusions were reached. I've provided line-specific examples below, but the authors may want to check for this throughout. Readability could also improve if the authors indicate biomarker interpretations directly in Figures 3 and 4 (e.g., label "Terrestrial input" next to sterols).

We appreciate the reviewer's suggestions for improving readability as detailed in our responses to the specific comments below. Following a suggestion by another reviewer, we also tried to guide the reader through our complex Figure 3 by adding arrows and interpretation on the figure (see revised version below).

[Figure]

In addition, the age unit is given in years in all figures but in ka in the text—please make this consistent. It would also aid comparison if Figure 4 panels were stacked vertically rather than side-by-side, and if time intervals were shaded in color to highlight different Holocene intervals.

We homogenized the ages to ka in the text and figures.

We partially adapted figure 4 to the reviewer: we aligned the records into 2 columns, and instead of shading different time periods, we added the time periods above the stacked figures (similar to Figure 3).

[Figure]

Low          Intermediate          High

PIP$_{25}$

Sea-ice thickness (m)

1.5

0

ARAC04    ARA2B
PCB11
PCB9

PS2458
PS51-154
PS51-159

90°E

85°N    BP00-36

75°N

Ryder19-12GC

Late          Middle          Early

**PS51-154**

IP$_{25}$ (µg/gOC)

PIP$_{25}$

Late          Middle          Early

**PS2458**

IP$_{25}$ (µg/gOC)

PIP$_{25}$

**ARA2B**

IP$_{25}$ (µg/gOC)

PIP$_{25}$

**PS51-159**

IP$_{25}$ (µg/gOC)

PIP$_{25}$

**ARAC04**

IP$_{25}$ (µg/gOC)

PIP$_{25}$

**BP00-36**

IP$_{25}$ (µg/gOC)

PIP$_{25}$

IP$_{25}$ (µg/gOC)
PCB11
PCB9

PIP$_{25}$
PCB11
PCB9

**Ryder-12GC**

IP$_{25}$ (µg/gOC)

PIP$_{25}$

Age (ka)          Age (ka)

Specific comments

L20: "…IP25, and other HBIs,…" → suggested phrasing "HBIs including IP25."'

We revised the text accordingly

L42: "…renewed…" needs context—did the interest lapse previously?

We think that the topic has attracted more attention in the last decade when the absence of sea ice in the Arctic was forecasted. However, we see the point of the reviewer and deleted that term.

L56: Add reference.

The references have been added L56-58 "*Numerous studies on Arctic sea ice variability have focused on offshore locations highlighting heterogeneity in sea-ice cover history and the importance of local currents (Belt et al., 2010; Detlef et al., 2023; Fahl & Stein, 2012; Hörner et al., 2016, 2018; Stein et al., 2017; Stein & Fahl, 2012; Vare et al., 2009; J. Wu et al., 2020).*"

L58: Use "Lipid biomarkers" for consistency with the title and throughout.

The text was changed L70 and throughout.

L62–63: "Several proxies for sea temperature exist using microfossils…" reads awkward; please rephrase.

The sentence was rephrased L74-77 "*In contrast to salinity, several established proxies exist for reconstructing sea temperature, including microfossil assemblages (e.g., dinocyst, Richerol et al., 2008), inorganic ratios (e.g., Mg/Ca of foraminifera, Barrientos et al., 2018; Kristjánsdóttir et al., 2007) and lipid biomarkers (Ruan et al., 2017; Varma et al., 2024).*"

L65–66: The phrase "usually include hydroxylated…" is too strong; OH-GDGT proxies are not yet widely used enough for "usually."

For the case of polar environments, they have been used for the last 10 years (since the discovery of Liu et al., 2012) which is, in our opinion, long enough to warrant the term "usually".

L80: "Material" → "Materials."

This has been changed in the revised text.

L83: Include and label Mackenzie Trough in the map for non-expert readers.

We labeled the Mackenzie Through in the revised Figure 1, see below.

[Figure]

L94: "…an open water flaw leads occur…" — if singular, use "occurs"; if plural, remove "an."

The "s" was moved to now read "an open water flaw lead occurs".

L99–102: Clarify whether multicores were used to generate proxy records. If not, remove.

The top slice of the multicore was used and this will be mentioned in the revised text (L117-119).

L103: "…as in (Matsuoka et al., 2012)." → "…as in Matsuoka et al. (2012)." Check and correct similar formatting errors throughout.

The formatting was checked and corrected throughout.

L110: "extend" → "extent."

This has been changed in the revised text.

L113: "…scanned shipboard on a Geotek…" → reword for clarity.

We disagree with the reviewer as this is common writing for this instrument.

L121–123: Add more explanation of the assumptions underlying these interpretations.

We added more details on the use of these ratios L138-139 *"Zr/Rb was used as a proxy for grain size variations as Zr content is elevated in coarse mineral, while Rb is associated with clay minerals (L. Wu et al., 2020) and Br/Cl as a proxy for marine organic matter as Br usually correlates with OC content (Wang et al., 2019)."*

L132–133: Define "well-preserved"; consider adding SEM images of foraminifera.

We added images taken on a binocular in Figure S6.

L156–157: Rephrase to clarify that 5 g refers to each depth/sample, not the whole study.

*We added L174-175 "For each sample, 5 g of homogenized freeze-dried sediment was extracted using an Energy Dispersive Guided Extraction (EDGE)"*

L165: C46 GTGT

*We corrected to L184 "and $C_{46}$ GDGT-like compound"*

L170: "Concentration of IP25 were…" → "was."

*This has been changed to "Concentrations of $IP_{25}$ were" in L191.*

L173: The PTFE filter should likely be 0.45 µm, not 45 µm. Please check.

*It has been corrected in L194.*

L175: "according to Hopmans et al. 2016 and following Lattaud et al. 2021" → clarify which method was actually used.

*The Lattaud et al. 2021 method was used, which is based on Hopmans 2016. This was revised in the text L196-197.*

L176: C46 GTGT (check notation).

*L197-198 "GDGTs were quantified using the $C_{46}$ GTGT internal standard assuming the same response factor."*

L177: "sililated" → "silylated."

*Corrected accordingly.*

L178: "C22 5,16" → add "-diol."

*Added accordingly.*

L180: Define "IRMS" at first mention.

*We added the definition of the acronym (L203).*

L208–209: Use consistent terminology for "salinity" vs. "S."

*We changed it in the equation accordingly.*

L211: Define "SST" here, not later (L216).

*We defined SST upon first use (now L234).*

L218–219: "SST" is italicized in L218 but not L219—be consistent.

We removed the italic formatting throughout.

L227: Only N. pachyderma counts are presented, yet the methods mention all species. Please clarify.

Yes, only *N. pachyderma* is presented in the manuscript as this was the only species we found. However, we looked also for other species that could have been present, so the method description is open.

L241–243: Plot sedimentation rate in Figure 2 for clarity.

We do not think sedimentation rate are needed on top of the age model in Figure 2 and prefer to keep the Figure as is.

L248–254, 257–259: These sentences read more like discussion than results—consider moving or rephrasing.

We removed it here as this is already mentioned in the discussion.

L260–262: Add figure citations so readers can locate corresponding results.

We added the figure citation to this paragraph.

L273–274: See General Comment [3]. It's difficult to locate data at 1 ka; consider adding minor ticks and grid lines.

We added grid lines in Figure 4 and 5.

L285: Consider moving Fig. S3 to the main text (only 5 figures currently).

We think Figure S3 should remain in the supplement as it mainly deals with the age model and the geochronology of the study cores is not the main focus of the paper. We will rather consider Figure S4 which deals with the core-top calibration of two of the most novel proxy for salinity and temperature.

L301–308: Same as above. Alternatively, briefly state in Methods that RI-OH′ was not used because it yielded unrealistic SSTs.

We would like to discuss the data and state that the RI-OH′ yielded unrealistic SST, and therefore prefer to keep it in the results section.

L349–351: The results do not clearly show that SST follows summer insolation. To me, SST declines at the end of the Early Holocene while PIP25 remains low, suggesting weak coupling.

We only claim that SST were elevated at the early Holocene, not that it was tied to insolation during the whole Holocene L391-393: "*During 12 – 8.5 ka, SST are elevated in comparison with the rest of the Holocene (Fig. 4d) which coincided with peak 21 June insolation*"

L363–364: This interpretation assumes Ti is relatively constant—please justify or provide supporting evidence.

Ti counts are stable throughout the core, we added all XRF data into a supplementary Table S3.

L365: "slop" → "slope."

Corrected accordingly.

L372: "have" → "has" (subject is singular).

Corrected accordingly.

L377–378: The conclusion "stable sea ice-edge or polynya conditions" is not well-supported—please elaborate.

At 7-6 ka PIP25 in PCB9 is above 0.5 indicating marginal sea ice zone conditions. We rephrased L419-422 "*At PCB09, SSTs cooled from ~6 °C to 3 °C (Fig. 4d), while steadily increasing sea-ice biomarker concentrations led to $PIP_{25} > 0.5$ by 7-6 ka, indicating the establishment of stable ice-edge or polynya conditions at the slope. The higher salinity (Fig. 4b) and greater distance from the coast at this site suggest enhanced influence of offshore Pacific-derived waters and reduced terrestrial input.*"

L386–400: See General Comment [3].

See our earlier response.

L413: "close by" → "close-by"

Corrected accordingly.

L417: "Norther" → "Northern"

Corrected accordingly.

L440: "Conclusion" → "Conclusions"

Corrected accordingly.

L451–453: This claim is not supported by the lack of correlation between the PIP25 and SST records.

We build our claim on the observation of the higher SST during the Holocene in link with the absence of sea-ice cover in all marginal seas.

Overall assessment:

This is a valuable contribution to Arctic paleoceanography, and I commend the authors for assembling such a comprehensive dataset and transparency reporting reporting all results,

including proxies that did not yield clear signals. Addressing the points above—particularly regarding biomarker interpretation, temporal resolution, and figure clarity—will greatly improve the robustness and accessibility of the paper.

We appreciate this positive assessment of our study and the constructive criticism. With this revised version, we hope to have satisfactorily addressed the reviewer's concerns.

---

## Author Comment (AC3)

**Answer to Reviewer 2's comment:**

Santos et al. present two new multi-proxy paleoceanographic records from the eastern Beaufort Sea. The unique sampling sites and wide range of proxies analyzed have the potential to enhance our understanding of Holocene sea-ice regime in this underexplored region. However, the current version suffers from methodological inconsistencies and a lack of focus in presentation (e.g., trying to do both analysis of site-specific and pan-Arctic trends), which limit its suitability for publication at this stage. While many proxies are analyzed, few are discussed in sufficient depth. The authors are encouraged to narrow their focus to the regional paleoceanography of the Beaufort Sea and to compare their results more thoroughly with nearby cores - especially Wu et al., 2020, which employed a very similar set of proxies. It is also important to critically evaluate the suitability of each proxy for this region. Indices such as RI-OH′ are still relatively new, and their environmental interpretations should be treated with appropriate caution until further validation is available. Below are more detailed suggestions to help strengthen the manuscript.

We thank the reviewer for their thoughtful and constructive comments. We will respond point-by-point below (in blue).

We acknowledge that some methodological details and proxy comparisons require clarification and we will revise the methods section to ensure consistency and transparency.

Concerning the use of recently developed proxies (e.g., RI-OH′, TEX-OH, $\delta^2H$ of $C_{16}$), we agree that their interpretation should be approached with caution. However, given the scarcity of validated Arctic-specific proxies, testing and cross-evaluating these indices remains valuable. We will clarify their potential and limitations more explicitly in the revised discussion.

We appreciate the reviewer's suggestion to narrow the focus of the study. We agree that the regional perspective is central, and we will ensure that the discussion clearly emphasizes the Beaufort Sea record. At the same time, we believe that retaining a concise Arctic-scale overview adds essential context. Regionally, our coastal core provides new insight into the Holocene sea-ice history of the Beaufort Sea, complementing the offshore record of Wu et al. (2020). To our knowledge, this is the first coastal record available for this region. Placing these results within a broader Arctic framework allows us to illustrate how local conditions relate to large-scale climate forcing, such as variations in insolation, ocean circulation, and bathymetry, which together shaped spatial differences in sea-ice cover during the Holocene.

INTRODUCTION

Overall, this introduction is brief, glances through many proxies without properly explaining their mechanism, limitations, and justifying their applicability to your specific study area. It also doesn't establish well the theoretical link between numerous oceanographic conditions and sea ice. The authors fail to identify knowledge gaps in the existing Holocene sea ice literature. If Holocene sea ice records have no regional heterogeneity among them, there is then no need for another paleo sea ice reconstruction. I understand the purpose of the statement in lines 55-56, but this present study does not offer more cores or a more coastal location than other average paleo sea ice studies. Regarding line 68-69 identified a valid gap in regional calibration for GDGT proxies, and this led naturally to the need for calibration from surface sediment samples, which was briefly mentioned in line 74-75, but this

connection is textually hard to see. The authors should put more effort into highlighting this connection and use citations to support their claim.

*We amended the introduction L55-58 to highlight the difference in sea-ice history reconstruction in the Arctic "Numerous studies on Arctic sea ice variability have focused on offshore locations highlighting heterogeneity in sea-ice cover history and the importance of local currents (Belt et al., 2010; Detlef et al., 2023; Fahl & Stein, 2012; Hörner et al., 2016, 2018; Stein et al., 2017; Stein & Fahl, 2012; Vare et al., 2009; Wu et al., 2020).".*

*We disagree with the statement that our study does not offer more core/more coastal location. The strength of this study is to analyse both cores (on the slope and on the shelf) which has only been done in one other location in the Laptev Sea. To add, most of these aforementioned Holocene sea-ice reconstructions from the Arctic have been derived from offshore settings, where records predominantly reflect pack-ice dynamics. However, comparatively few records exist from coastal or inner-shelf environments, where landfast ice and seasonal polynya activity exert dominant controls on sea-ice conditions.*

*Regarding the proxies, we added L81-83 "These limitations highlight the need to further develop and test Arctic-specific proxies for both salinity and sea temperature."*

line 47-51: As one of your cores covers the end of the last deglacial, discussing some pre-/early Holocene warming events that you expect to see in your record in chronological order might help streamline your narrative.

*We currently describe cooling events in chronological order but not the warm events pre-Holocene, so we amended the text L49-52: "Throughout the Holocene, Arctic sea ice has responded to changes in orbital forcing, ocean circulation, and ice sheet dynamics (Park et al., 2018; Stein et al., 2017). During the last deglacial, abrupt climatic events such as Bølling-Allerød (~14 – 12.8 ka) and Younger Dryas (~12.8–11.7 ka), contributed to the instability of the Arctic cryosphere. In the Canadian Arctic, the enhanced meltwater discharge and re-routing following the retreat of the Laurentide Ice Sheet (LIS) contributed to oceanographic shifts and transient cooling events (Broecker et al., 1989)."*

line 52-54: The summary of Holocene sea ice trend in line 52-54 might be too simplistic and overlook the regional disparity among sea ice records. The narrow age constraint on the Holocene thermal maxima is questionable. If your claim only describes the Beaufort Sea, please specify that, as it has not been made clear.

*We go into more details about spatial heterogeneity of Arctic sea-ice records in the last paragraph of the discussion.*

*Thanks for pointing out that the timing of the HTM is too narrow, we updated the time period for the western Beaufort Sea. We now followed Kaufman et al. 2004 timing and revised for 11 to 6 ka.*

line 55-56: Which studies? Need citation or need to reformulate the sentence.

*We added these references L57-58.*

line 57-69: The authors presented the current arctic paleoceanography proxy toolbox in a confusing order, while leaving out the key HBI-based proxies and index (IP25, PIP25), which definitely deserve some discussion. There are numerous GDGT-based paleothermometers and indices; the authors should name them directly in the paragraph to avoid confusion. Please consider adjusting the use of "e.g." from line 60 onward, especially for line 63. Regarding line 69 needs citations to support the claim.

*We tried to limit the use of "e.g." when citing references when possible. We added one paragraph on sea-ice proxies including IP25 and PIP25 L60-68 "Sea-ice cover can be reconstructed from microfossil and lipid biomarker evidence preserved in marine sediments. Remains of sea-ice organisms such as dinocysts (de Vernal et al., 2013) and diatoms, the latter producing a specific biomarker known as $IP_{25}$ (Belt et al., 2007), provide valuable records of past sea-ice conditions. This highly branched isoprenoid (HBI) and its isomer HBI diene (HBI-II) are used to trace the presence of spring sea-ice in modern and geological settings. However, because the absence of these two HBIs may reflect either a permanent sea-ice condition (due to the absence of light) or completely sea-ice free waters, the $PIP_{25}$ ratio was developed (Müller et al., 2011). This ratio includes a phytoplankton biomarker (typically dinosterol, brassicasterol or HBI-III) that represents open-water productivity. $PIP_{25}$ values have been used to distinguish between seasonal sea-ice (>0.5) and permanent sea-ice cover (>0.75).".*

*For the other proxies. we follow the order stated in L60 "salinity, sea temperature and freshwater influence". Specifically, for sea temperature proxies the order is historical with first, the description of microfossil use (here we use e.g. as this is not the focus of the study), the inorganic ratio (again e.g. as this is not the focus), ending the list with biomarkers. We spelled out the names of the GDGT ratios used in SST reconstruction L77-79 "Among biomarker proxies for cold water (< 15°C) environments, hydroxylated glycerol dialkyl glycerol tetraether (OH-GDGT) are particularly useful, with RI-OH' and TEX-OH identified as promising temperature indices (Lü et al., 2015; Varma et al., 2024)".*

line 70-79: The objectives 2 and 3 are bold statements; some reformulation while keeping the limitation of your proxies in mind is needed.

*We rephrased the objectives L90-93 "The primary objectives are to (1) reconstruct past variations in sea-ice cover on the Beaufort Shelf throughout the Holocene, (2) explore the potential roles of insolation changes, meltwater input, and oceanic conditions in shaping regional sea-ice variability, and (3) place the Beaufort Sea record within a broader Arctic context to provide insights into past and present climate variability."*

MATERIALS AND METHODS

Overall, the biomarker workflow is unconventional. The authors attempt to analyze too many biomarker classes from a single extract, which may compromise the analytical quality of each individual proxy. The solvent system, choice of standards, and selected m/z values raise concerns about data robustness. Methodologically, the section reads as a workflow appears fragmented and could benefit from clearer structure with uneven detail across proxies. While I appreciate the challenge of condensing complex workflows into a limited space, a clearer structure, such as a summarized workflow diagram or table, would greatly help readers follow the analytical sequence and evaluate reproducibility.

We disagree with the reviewer on the unconventional workflow, these protocols have been used in many other studies (see the interlaboratory comparisons of Belt et al., 2014; Bijl et al., 2025; De Jonge et al., 2024). The solvent system for solid-phase chromatography separation is ideal to separate alkenones from the rest of the lipids (in the DCM fraction). We agree that the choice of standard for the sterol is unusual and will not give a perfect quantification due to their different structure, this is also why we do not focus on absolute concentrations but rather the relative difference within the sterols. The sterols are not used to calculate the PIP25 or other ratios where this could be an issue.

It is unusual to have a table of workflow in such manuscripts, especially for established methods following interlaboratory recommendations (Belt et al., 2012, 2014), (Bijl et al., 2025; De Jonge et al., 2024).

Figure 1: Where are the surface sediment samples? Please consider using a legend for the labeling of sea ice extent. Country names are not necessary; authors should use a brighter color to indicate the study area in the global map, or else the small map will not be very useful.

The surface sediment samples are presented in Figure S1 as they are not part of the main discussion and serve as support for some of the newer proxies used in this study. This is mentioned L117-119 "*The core tops (0-1cm) from 22 multicores collected during PeCaBeau, were used to ground truth the hydrogen isotope ratio of $C_{16:0}$ fatty acid proxy for reconstructing salinity and test the applicability of the temperature reconstructions (Fig. S1).*"

We will remove the country names and change the color for the study area in the revised figure 1 (see revised figure below).

[Figure]

Figure 1

line 156: Could you clarify what is the resolution of your biomarker analysis?

We added one sentence to the revised text to define the number of samples and resolution in the core L173-174: "*Lipid biomarkers were analysed from 42 samples (every 10 cm for the*

*first 143 cm, then every 20 cm) for PCB09 and 21 for PCB11 (every 20 cm). Core top samples from the MC's were also analysed for lipid biomarkers.".*

line 158: Could you clarify what is the concentration and the composition of the alkaline solution used for saponification?

We will add this detail in the revised text L178 (stated in the reference study Lattaud et al., 2021), we used a KOH in methanol solution at 0.5M.

line 163: Could you explain why internal standards added post-extraction?

We acknowledge that it would have been better to add the internal standard before extraction to assess loss during laboratory work and to assess extraction efficiency. However, most losses occur during workup after extraction so the effect of adding the internal standards after extraction only add a small uncertainty.

line 164: The choice of C22 5,16-diol as an internal standard for sterols is unconventional. As a long-chain diol, its polarity, structure, and chromatographic behavior may differ substantially from sterols, and it is unlikely to mimic sterol recovery or derivatization efficiency.

We agree that is it unconventional for this standard to be used for sterol quantification, however it is still a complex alcohol molecule. Here we do not stress the absolute concentration of the sterols in our records but rather the relative changes.

line 170: HBI IV is an isomer of HBI III with the same degree of unsaturation. Why is it monitored at 348? The authors should include ion monitoring values for the standards as well, since previously there have been a few different fragments monitored for the same standard (For example, 7-HND can be monitored with m/z 99 and 266).

We added the ion monitoring values for the standards to the revised methods, here 266 for 7-HND and 350 for 9-OHD. We agree that HBI IV and HBI III are monitored at the same m/z, this was a writing mistake, which is now corrected in the revised text L192.

line 172: Please provide comparison data in the supplementary material.

We will provide the concentration of the reference material as supplement (Table S2).

line 173: Saponification is not a viable strategy for GDGTs work. Assuming your post-saponification liquid-liquid extraction is between a methanolic KOH and hexane:DCM mix, there is a possibility that quite some GDGTs will be lost in the aqueous phase.

We disagree with the reviewer here. Saponification is a viable strategy for GDGT analysis as was shown in the latest recommendation for handling GDGT (Bijl et al., 2025). It has also been shown to be the ideal method to quantify sterols (Fu et al., 2025).

line 177: This raises concerns and warrants clarification if the sterols would end up in F3, as their polarity is different than GDGT and according to this protocol, would probably mostly elute in F2 (DCM) and potentially partially elute in F1. Also, how is the F3 used both for GDGTs and phytosterols? Is it a split of the fraction? This point please clarify.

Previous protocols for GDGT analysis (Bijl et al., 2025, Lattaud et al., 2021) show that GDGTs end up in fraction 3. No sterol eluted in F1 or F2 which were monitored for other compounds (for another project these fractions were screened for alkanes and alkenones). F3 was split in 2 with one fraction analysed for GDGTs and one fraction ran for sterol analysis. This is added in the revised methods L194.

line 178: What are the m/z ratios, and which sterols are you quantifying?

We used the total ion current for quantification, m/z = 129 and specific ions were used for identification of brassicasterol, stigmasterol, β-sitosterol, campesterol, see revised L202.

line 203: c factor should be reported here.

Initially the c factor was only in the supplementary, we added it to the revised method section L227.

line 202: The statement of not being able to detect dinosterol in the samples is concerning, since dinosterol should be a regionally abundant biomarker. In Wu et al. 2020, the core, which is nearby, the PIP25 index was calculated solely based on dinosterol. The complete absence of dinosterol raises questions about the sterol recovery in this study under review.

Dinosterol has been absent from other Beaufort Sea studies (Belt et al., 2013; Fu et al., 2025) and its absence in our study sites is expected. Instead, its presence in Wu et al. site is surprising and could be due to local currents bringing different amount of nutrients. We now add this information L225"*Dinosterol was not detected in the samples which is common in the Beaufort Sea (Fu et al., 2025),*"

RESULTS

Figure 3: The practice of including brassicasterol as a terrestrial sterol is potentially problematic and warrants reconsideration, even regionally, brassicasterol is primarily of terrestrial origin, but it's still a sterol that has a mixed source.

We agree with the reviewer that brassicasterol needs to be interpreted with caution and can for our study region be used as a primarily terrestrial biomarker. This is indicated L224-225: "*brassicasterol has been shown to derive mainly from terrestrial input in the region* (Wu et al., 2020)."

Figure 3&4: The authors should consider grouping proxies that are reconstructing the same information in the same figure. (such as PIP25 with the HBIs, BIT with terrestrial sterols)

For figures 3 and 4 we prefer to present all biomarker concentrations in figure 3, and the biomarker ratios in figure 4.

DISCUSSION

line 335: The authors claim that the comparison between the slope core and outer shelf core is the focus of this study, but the comparison is weak throughout the discussion. For most of the discussion, the authors seem to treat both cores as one single record without emphasizing the difference in their depositional environments. In the introduction (line 74) and results (line

289), the authors claim that they conducted further calibration work with surface sediment, but these samples weren't included in the map, nor do we see the data presented or discussed anywhere else.

We appreciate the reviewer's suggestion to clarify the comparative aspect between the slope (PCB09) and outer shelf (PCB11) cores. We would like to note that our comparison is constrained by the temporal coverage of the records: the Deglacial–Early Holocene interval is only preserved at PCB09, preventing a direct comparison for this period. However, for the overlapping Middle–Late Holocene interval, we explicitly highlight the spatial contrasts between the two sites, for example: differences in the timing of sea-ice stabilization (PCB09 at 7–6 ka versus PCB11 at 4 ka), the distinct depositional settings (flaw-lead versus stable ice-edge), and associated productivity trends. We rewrote part of the discussion to emphasize this part of the study (see paragraph 4.1. and 4.2.).

The core-top sediments are presented in Figure S1 and the results for the calibration of the salinity proxy are described L319-324 and reported in figure S4. We added some details on the SST reconstruction in the surface sediment in the revised text L338-340 "*RI-OH' in the surface sediments varies from 0.05 to 0.17 while TEX-OH varies from 0.08 to 0.32. Both indexes plot in the global calibration curves from (Varma et al., 2024) and the reconstructed SST varies from 0.9 to 4.0 °C and -0.1 to 11.6 °C, respectively.*", we also added a figure for the calibration of the SST indexes (see below).

[Figure]

Figure S4

line 337: The use of HBI II as an arctic paleo sea ice proxy is far less common than HBI I; it does not add anything to the author's narrative. It is redundant and adds to the confusion. The same goes for HBI IV, since this paper isn't exploring the HBI TR25 index, there is no real reason to present the HBI IV as a separate record.

We do not agree with the reviewer, although new, the use of HBI-II can bring additional information for future studies and as such is included in our results. Its behaviour in both cores reflects clearly IP25 (Fig. 4) hence we do not find it confusing but rather a valuable

information. Moreover, previous studies have identified HBI-II to be at least a useful alternative when IP25 is absent or falls below detection limits (Andrews et al., 2018).

line 345: Please consider reformulating "some sea ice coverage".

We changed it for "intermittent" L387.

line 348: Could you clarify what is the reasoning behind this heterotrophic production claim? Citations are needed.

Here we use isoGDGT as a tracer for ammonia -oxidizers as mentioned L389-390 "*Heterotrophic production in the shelf slope region during this period is relatively low (as suggested by the presence of ammonium oxidizer Thaumarchaea-derived isoGDGTs*". We added a reference for the production of GDGT by Thaumarchaeota.

line 399-400: This claim needs some further support.

We reformulated for clarity L441-444 "*In contrast, on the outer shelf, seasonal sea-ice conditions persisted longer and sea ice cover expanded gradually and became well established after about 2 ± 0.6 ka. Even as sea-ice biomarkers increased, open-water diatom markers remained relatively abundant, implying continued flaw-lead or marginal-ice-zone productivity sustained by intermittent open-water formation and coastal influence.*".

Figure 5: All records should have a shared time axis. If authors insist on presenting both IP25 and PIP25, they should separate them into two clear columns.

We adapted the time axis to make sure aligned records share the same axis. We do not think separating IP25 and PIP25 more than it is now done (see revised figure below) is needed.

[Figure]

Figure 5

line 450: The resolution of biomarker analysis in this study doesn't allow the authors to make claims about centennial events like the Little Ice Age.

We rephrased this part in the revised text to reflect the reviewer's comment L439-440 "*These changes are broadly consistent with the timing of the regional cooling associated with the Little Ice Age (Mann et al., 2009), though the resolution of the biomarker record does not allow precise attribution to centennial-scale events.*".